# Dynamics and nanoscale organization of the postsynaptic endocytic zone at excitatory synapses

**Lisa AE Catsburg, Manon Westra, Annemarie ML van Schaik, Harold D MacGillavry***

Cell Biology, Neurobiology and Biophysics, Department of Biology, Faculty of Science, Utrecht University, Utrecht, Netherlands

**Abstract** At postsynaptic sites of neurons, a prominent clathrin-coated structure, the endocytic zone (EZ), controls the trafficking of glutamate receptors and is essential for synaptic plasticity. Despite its importance, little is known about how this clathrin structure is organized to mediate endocytosis. We used live-cell and super-resolution microscopy to reveal the dynamic organization of this poorly understood clathrin structure in rat hippocampal neurons. We found that a subset of endocytic proteins only transiently appeared at postsynaptic sites. In contrast, other proteins were persistently enriched and partitioned at the edge of the EZ. We found that uncoupling the EZ from the synapse led to the loss of most of these components, while disrupting interactions with the actin cytoskeleton or membrane did not alter EZ positioning. Finally, we found that plasticity-inducing stimuli promoted the reorganization of the EZ. We conclude that the EZ is a stable, highly organized molecular platform where components are differentially recruited and positioned to orchestrate the endocytosis of synaptic receptors.

**\*For correspondence:**
h.d.macgillavry@uu.nl

**Competing interest:** The authors declare that no competing interests exist.

## Editor's evaluation

The delineation of protein organization in the perisynaptic endocytic zone is an important contribution to our understanding of synapse structure, and new observations about changes to this structure establish intriguing new phenomenology that appears closely linked to synapse functional plasticity. Cutting-edge genetic tagging and elegant application of super-resolution imaging compellingly support the key claims in the paper.

## Introduction

Clathrin-mediated endocytosis is the principal mechanism for the internalization of membrane components, and is essential for cellular homeostasis, intercellular signaling, and nutrient uptake in mammalian cells (*Kaksonen and Roux, 2018*; *McMahon and Boucrot, 2011*; *Mettlen et al., 2018*). This process involves the tightly controlled initiation and maturation of clathrin-coated pits that is mediated by the sequential recruitment of clathrin, cargo, and endocytic adaptor proteins (*Cocucci et al., 2012*; *Taylor et al., 2011*). Apart from these well-characterized, small (~100 nm) and short-lived ( < 120 sec) clathrin coats, numerous electron and (live-cell) light microscopy studies have revealed that clathrin can assemble into a remarkably large variety of membrane-attached structures (*Grove et al., 2014*; *Heuser, 1980*; *Leyton-Puig et al., 2017*; *Saffarian et al., 2009*; *Sanan and Anderson, 1991*). In fact, the lifetime, size, and morphology of clathrin assemblies at the membrane diverge enormously between cell types and even within cells. Clathrin structures varying from 100 nm up to 1 μm with

lifetimes ranging from seconds to tens of minutes have been reported. The origins and functional relevance of this striking heterogeneity remain to be elucidated.

This heterogeneity is particularly evident in neurons that contain a divergent population of clathrin structures distributed over their immense and complex plasma membrane. At postsynaptic sites, a clathrin-coated structure (CCS) referred to as the endocytic zone (EZ) is stably associated with the postsynaptic density (PSD) (*Blanpied et al., 2002*; *Lu et al., 2007*), via a Shank-Homer1c-Dynamin3 interaction (*Lu et al., 2007*; *Rosendale et al., 2017*). Disrupting the PSD-EZ interaction severely affects glutamate receptor levels at synapses. Particularly, the ionotropic AMPA-type glutamate receptors (*Petrini et al., 2009*; *Rosendale et al., 2017*) and metabotropic glutamate receptors (*Scheefhals et al., 2019*) have been found to undergo trafficking mediated by the EZ, while transferrin receptors are not preferentially internalized near the synapse (*Rosendale et al., 2017*). It has been proposed that once internalized at the EZ, glutamate receptors enter the local recycling mechanism, that retains receptors in intracellular pools that can recycle back to the synaptic membrane in an activity-dependent manner (*Park et al., 2006*). Indeed, the local recycling of receptors via the EZ is essential for synaptic plasticity as uncoupling the EZ from the PSD depletes synaptic AMPA receptors and aborts activity-induced trafficking of receptors to the synaptic membrane during long-term potentiation (*Lu et al., 2007*; *Petrini et al., 2009*). Importantly, disruptions in EZ structure and function have been associated with the development of neuronal disorders such as autism spectrum disorder and Parkinson's disease (*Cortese et al., 2016*; *Scheefhals et al., 2019*).

Despite the clear functional importance of the EZ for synaptic transmission and plasticity in neurons, the molecular organization and how this organization contributes to its function is poorly understood. In electron microscopy studies, CCSs have been observed within dendritic spines (*Petralia et al., 2003*; *Tao-Cheng et al., 2011*) at an approximate distance of 100–600 nm from the PSD, coinciding with an enrichment of endocytic proteins such as dynamin2 and AP2 (*Rácz et al., 2004*). However, fundamental information on the spatial distribution and dynamics of endocytic proteins relative to the EZ and how these proteins contribute to EZ organization is missing. Here, we resolved the spatial and temporal organization of CCSs in dendrites and spines using live-cell imaging and super-resolution microscopy. We found that the postsynaptic EZ contains a unique and stable assembly of endocytic proteins, that is highly organized at the nanoscale level and is reorganized during synaptic plasticity. Based on these findings, we propose that the EZ is a highly distinct clathrin-coated structure that operates as a preassembled platform for endocytosis of synaptic components to sustain efficient synaptic transmission and plasticity.

## Results

### Heterogenous morphology of clathrin-coated structures in dendrites

To visualize CCSs in mature cultured hippocampal neurons (DIV16-21), GFP-clathrin light-chain-A (GFP-CLCa) was co-transfected with Homer1c-mCherry as a marker of excitatory synapses. We found a large variety of CCSs distributed throughout the entire neuron (*Figure 1A*). In dendrites, a high density of clathrin structures was found in the shaft and the majority of dendritic spines contained a distinct EZ, defined as a clathrin puncta closely associated with the PSD (75% ± 5%), consistent with previous observations (*Blanpied et al., 2002*; *Lu et al., 2007*; *Scheefhals et al., 2019*). Importantly, labeling endogenous CLCa using a CRISPR/Cas9-based approach (*Willems et al., 2020*) resulted in comparable distribution of clathrin structures (*Figure 1—figure supplement 1*). To resolve CCSs in dendrites at high spatial resolution, we used stimulated emission depletion (STED) microscopy, allowing quantitative analyses of clathrin structure morphology (*Figure 1B*). Notably, STED resolved individual structures at much higher resolution than confocal, often resolving distinct substructures within clathrin patches that appeared homogenous in confocal microscopy (*Figure 1C*). PSDs associated with more than one clathrin structure were also observed (*Figure 1B and C*). We found that 59% ± 5% of the PSDs were associated with one clathrin structure, while 14% ± 2% and 5.4% ± 0.8% were associated with two or three clathrin structures, respectively (*Figure 1C*).

Next, we analyzed the morphology of dendritic clathrin structures and found a large range of sizes from as small as 0.01 $\mu m^2$ up to 0.43 $\mu m^2$ (*Figure 1E*). On average, the area of PSD-associated clathrin structures was lower, albeit not statistically different from the average area of clathrin structures found in the shaft (area clathrin structures in shaft: 0.045 ± 0.003 $\mu m^2$, spine: 0.038 $\mu m^2$ ± 0.002,

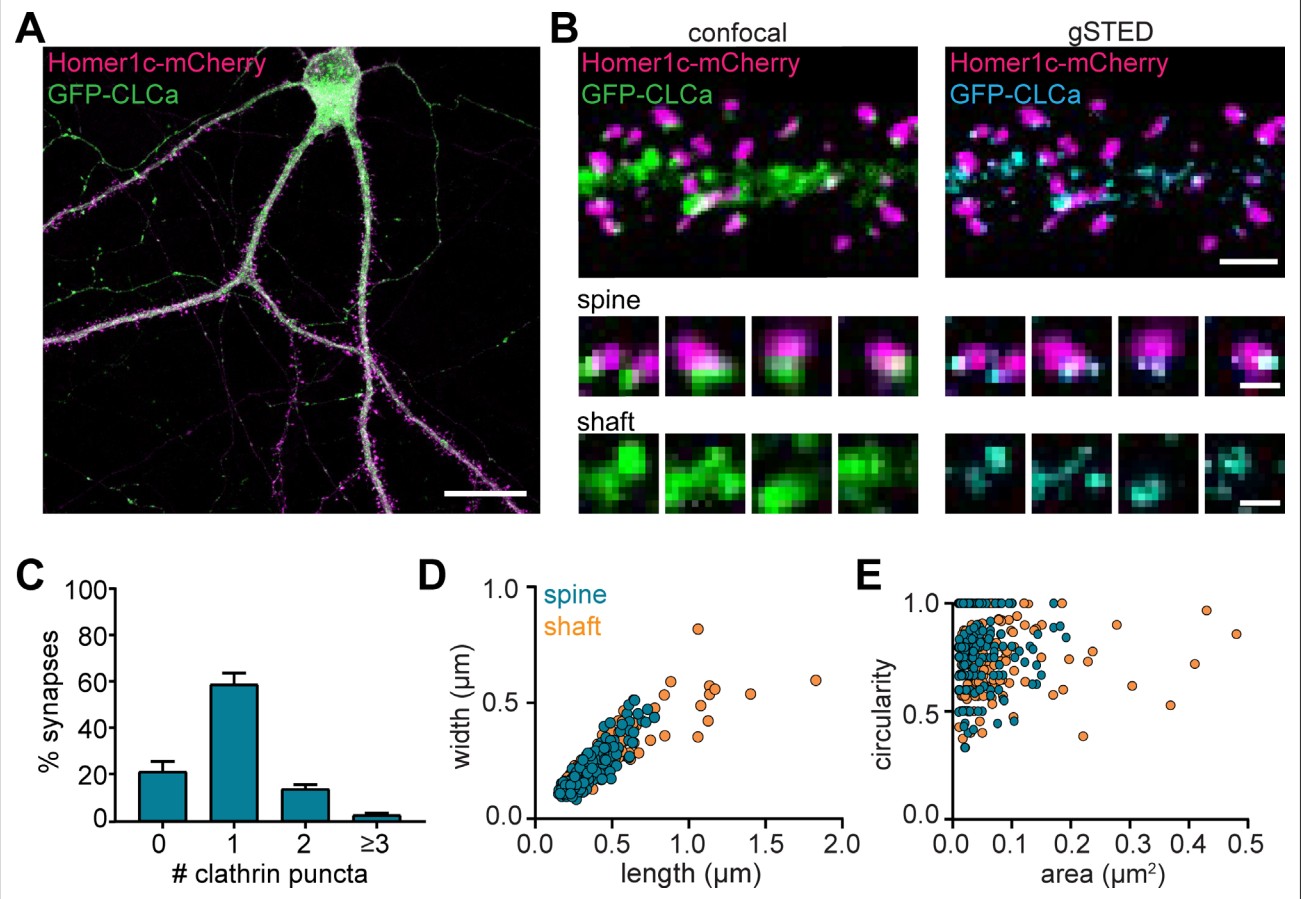

**Figure 1.** Heterogenous morphology of clathrin-coated structures in dendrites. (**A**) Example image of neuron expressing Homer1c-mCherry and GFP-CLCa. Scale bar: 20 μm. (**B**) Comparison of confocal and gSTED images of dendrite expressing Homer1c-mCherry and GFP-CLCa. Scale bars dendrite: 2 μm, zooms: 500 nm. (**C**) Number of CCSs per PSD per neuron, represented as mean ± SEM (N = 12 neurons). (**D**) Scatterplot of the length (μm) and width (μm) of CCSs in the dendritic shaft and associated with Homer1c based on ferret dimensions (spine: n = 248, shaft: n = 301). (**E**) Circularity ratio plotted against area (μm²) (spine: n = 248, shaft: n = 301).

The online version of this article includes the following source data and figure supplement(s) for figure 1:

**Source data 1.** Excel sheet with numerical data represented as plots in *Figure 1C, D and E*.

**Figure supplement 1.** Distribution of endogenously tagged CLCa in neurons.

**Figure supplement 1—source data 1.** Excel sheet with numerical data represented as plot in *Figure 1—figure supplement 1C*.

p > 0.1). However, the variability in sizes of clathrin structures in the shaft, was much larger than in spines (CV shaft: 1.3; spines: 0.84), with larger structures exclusively found in the shaft and not in spines (*Figure 1D and E*; range in width/length shaft: 0.15 μm to 1.7 μm, spine: 0.084 μm to 0.78 μm). Indeed, large clathrin structures were regularly observed in the shaft, approximately ~3 per 20 μm of dendrite (data not shown). Thus, dendrites contain a large variation of clathrin-marked structures, with PSD-associated EZs being a distinct, homogenous sub-population of clathrin structures in dendritic spines.

Studies in non-neuronal cells often classify clathrin structures as flat lattices based on size and shape. Compared to small, circular clathrin structures, presumably representing endocytic pits or intracellular vesicles, lattices are defined as large and irregularly shaped clathrin structures (*Grove et al., 2014*; *Leyton-Puig et al., 2017*; *Saffarian et al., 2009*). We generated scatterplots of the circularity and area of individual dendritic clathrin structures to test if we could find a similar classification (*Figure 1E*). However, there was no correlation between size and circularity in dendritic clathrin structures (R² = 0.006). Also, we did not observe a clear distinction in clathrin structures using this approach, suggesting that CCSs in dendrites form a highly heterogeneous population that, at this

resolution ( < 100 nm), cannot easily be classified based on these morphological parameters. Altogether, these data highlight the morphological heterogeneity of CCSs in neuronal dendrites.

## Clathrin dynamics at the EZ are distinct from CCSs in the dendritic shaft

To study the dynamic properties of CCSs in both spines and dendritic shaft we next performed live-cell imaging of GFP-CLCa in dendrites. We first investigated the dynamics of CCSs on short time intervals by imaging at 0.2 Hz for 5 min. To differentiate stationary from moving particles, we used a Fourier analysis-based filtering on kymographs (*Mangeol et al., 2016*). The dendritic shaft predominantly contained stationary CCSs, however smaller anterograde and retrograde moving puncta were also observed (*Figure 2A*). Interestingly, these fast-moving particles were small (IQR: 0.013–0.065 µm$^2$) most likely reflecting intracellular vesicle transport. Within the stationary pool in the shaft, we observed a few distinct CCSs. The two most frequently observed structures were larger, high-intensity structures that either remained fluorescently stable over the entire course of imaging (CV fluorescence intensity = 0.09, *Figure 2B*: upper panel), or showed large fluctuations in fluorescence intensity (CV = 0.16, *Figure 2B*: lower panel), perhaps indicating a more dynamic structure. In rare cases, transient budding of clathrin from these structures was observed, reminiscent of endocytic pit formation.

In spines, the EZ appeared much more stable than dendritic clathrin structures, with little fluctuations in GFP-CLCa intensity (CV: 0.02, *Figure 2C*: upper panel). Strikingly, we were able to pick up, what seemed to be the budding of individual vesicles from the EZ (*Figure 2C*, lower panel). On average, the fluctuations in intensity of clathrin structures in shaft and spines were significantly different, with much lower fluctuations found in spines (shaft CV: 0.06 ± 0.004, spine CV: 0.02 ± 0.001, p < 0.001) (*Figure 2D*). Longer acquisitions of 20 min at 30-s intervals showed that the CCSs in spines had a considerably higher average lifetime compared to CCSs in the shaft (average lifetime spines: 10 ± 0.7 min, shaft: 6.0 ± 0.3 minutes, p < 0.001) (*Figure 2E*). Indeed, 68.1% ± 6.0% of PSDs remained associated with at least one clathrin structure that was present for the entire 20 min, confirming that the EZ is stably coupled to the PSD (*Blanpied et al., 2002*; *Lu et al., 2007*; *Scheefhals et al., 2019*). In contrast, in the dendritic shaft, only a small fraction (~15%) of CCSs was long-lived ( > 17.5 min) and the median lifetime of all events was ~2.5 min, indicating that the majority of CCSs in the dendritic shaft are transient structures (*Figure 2E*).

The relatively long lifetime and small fluctuations in intensity of clathrin at the EZ might suggest a considerably lower turnover of clathrin at the EZ compared to shaft structures. To determine the turnover of clathrin at stable dendritic structures, we used fluorescence recovery after photobleaching (FRAP) of GFP-CLCa (*Figure 2F*). A relatively long baseline of 2 min was acquired to make sure that only stationary structures would be included in the analysis. We determined that in stable shaft structures GFP-CLCa recovered relatively fast (tau: 13.0 min) to 75.8% ± 10% in 20 min (*Figure 2G and H*), indicating a high level of clathrin exchange at these structures. In contrast to the high turnover of stationary structures in the shaft, the EZ showed relatively low levels of turnover (tau: 36.2 min) and total recovery (38.2% ± 5.2% after 20 min) (*Figure 2G and H*), suggesting little exchange of clathrin at the EZ. Taken together, these live-cell imaging experiments show that CCSs in the dendritic shaft are morphologically and dynamically highly diverse, and that the EZ in dendritic spines contains a stable accumulation of clathrin that is very similar from spine to spine, thereby differentiating itself from all other CCSs.

## Nanoscale organization of the endocytic zone in dendritic spines

To further resolve the spatial organization of the EZ, we next used single-molecule localization microscopy (SMLM). Homer1c-mCherry and GFP-CLCa were co-transfected as before and labelled with primary and secondary antibodies to perform two-color dSTORM imaging and reconstruct high-density localization maps of the distribution of clathrin molecules within the EZ and relative to the PSD (*Figure 3A*). We used DBScan to define clusters of Homer1c molecules, outlining the PSD and the associated clathrin clusters, marking the EZ (*Figure 3B*). We found that the centroid of the EZ was generally located within 100 nm from the border of the PSD (*Figure 3C*) with an average border-to-centroid distance from PSD to EZ of 10.6 ± 7.8 nm, confirming that the EZ is closely linked to the PSD and well within a distance that can be linked by scaffold proteins (*Lu et al., 2007*).

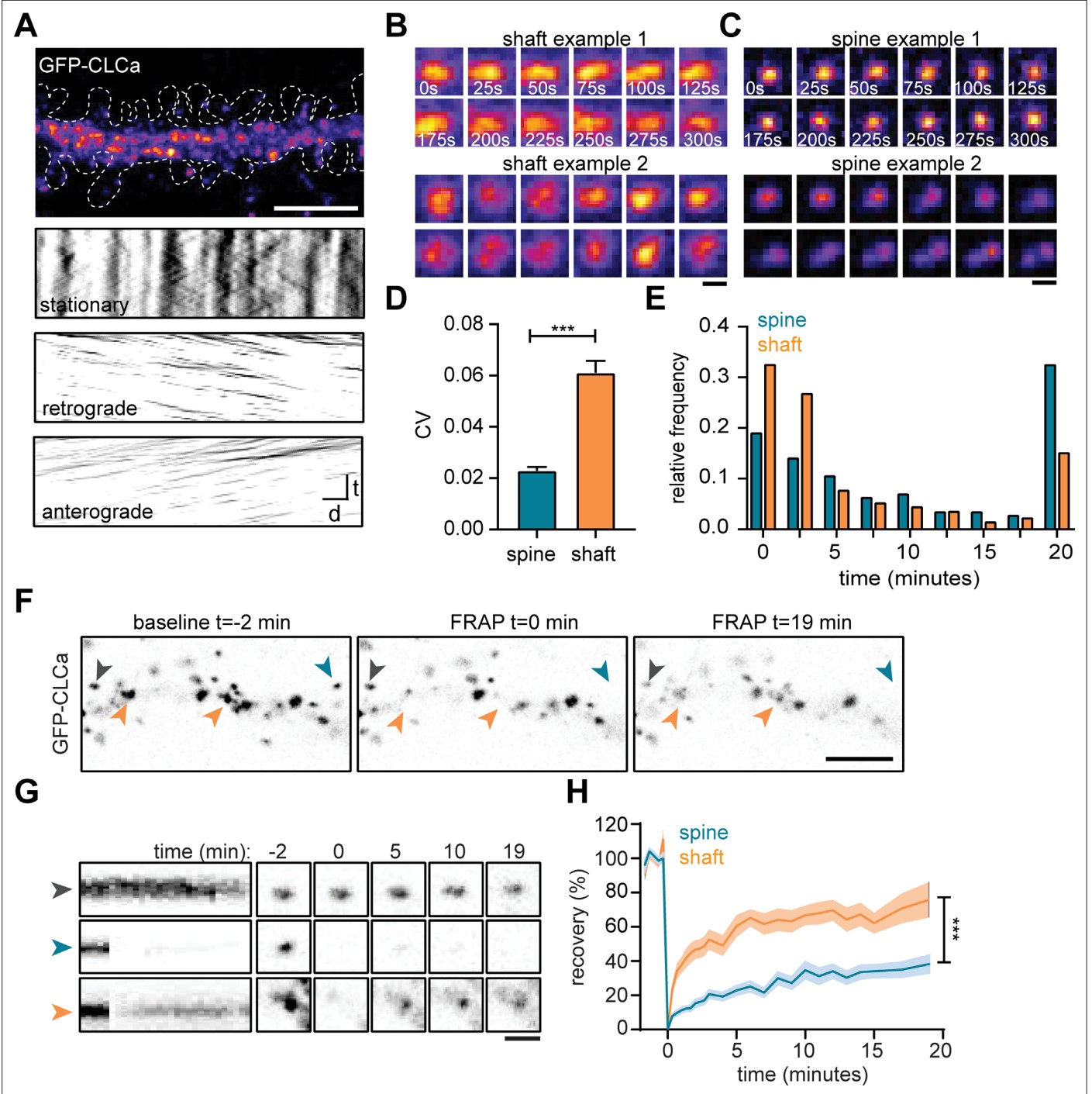

**Figure 2.** The EZ is dynamically distinct from shaft CCSs. (**A**) Representative dendrite expressing GFP-CLCa, scale bar: 5 μm, and kymographs of CCSs in the dendritic shaft only, separated in stationary (upper panel), retrograde (middle panel) and anterograde (lower panel) particles. Scale: time t, on the y-axis is 5 min, and distance d on x-axis is 20 μm. (**B**) Two examples of intensity fluctuations in stationary dendritic shaft structures. Scale bar: 1 μm. (**C**) Two examples of intensity fluctuations in spine structures. Scale bar: 1 μm. (**D**) Fluctuations in intensity plotted as the coefficient of variance (CV) between shaft and spine (spine: n = 48, shaft: n = 49, p < 0.001). Data represented at mean ± SEM. (**E**) Histogram of the lifetime of CCSs in shaft and spine (spine: n = 171, shaft n = 769), data represented as fraction. (**F**) Example images of GFP-CLCa before (left panel), directly after FRAP (middle panel) and recovery (right panel), scale bar: 5 μm. Gray arrow indicates control, unbleached region, blue indicated bleached EZ, orange indicates bleached stationary dendritic shaft structures. (**G**) Kymograph and example images of the structures indicated in F. Kymograph shows 22-min acquisition, scale bar: 1 μm. (**H**) Percentage of recovery in shaft (orange, n = 14) and spine (blue, n = 30).

The online version of this article includes the following source data for figure 2:

**Source data 1.** Excel sheet with numerical data represented as plots *Figure 2D, E and H*.

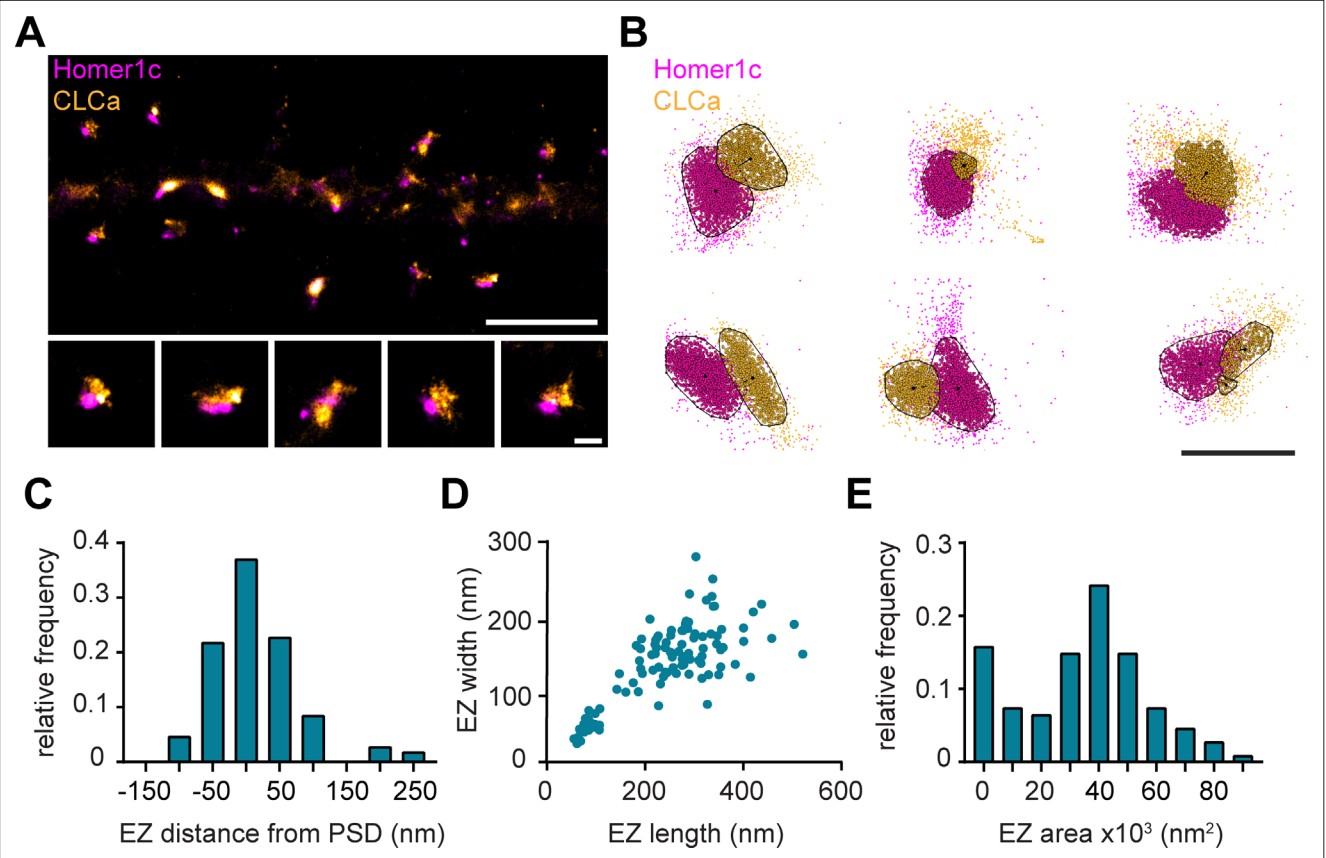

**Figure 3.** Nanoscale organization of the postsynaptic endocytic zone. (**A**) SMLM image of dendrite expressing Homer1c-mCherry and GFP-CLCa labeled with CF568 and A647 and zooms of individual EZs, respectively. Scale bar upper panel: 2 µm, zooms: 250 nm. (**B**) Individual molecules of Homer1c (magenta) and CLCa (orange) are outlined using DBScan. Black dot and line indicate center of the EZ (dot) and distance to the border of the PSD (line). Scale bar: 500 nm. (**C**) Histogram of the border (Homer1c) to center (CLCa) distance in nm. (**D**) Scatterplot of the FWTM length (nm) and FWTM width (nm) of the EZ. (**E**) Histogram of the area of the EZ plotted as x10³ nm². (**C–E**) n = 107.

The online version of this article includes the following source data and figure supplement(s) for figure 3:

**Source data 1.** Excel sheet with numerical data represented as plots *Figure 3C, D and E*.

**Figure supplement 1.** SMLM reveals nanoscale scale architecture of PSD-associated CLCa structures.

**Figure supplement 1—source data 1.** Excel sheet with numerical data represented as plots in *Figure 3—figure supplement 1B,C,D,E*.

On average the area of the EZ was 35.0 ± 2.0 × 10³ nm² (*Figure 3E*), and 224.6 ± 10.4 nm in length and 146.5 ± 5.2 nm in width (*Figure 3D*). Moreover, the dimensions (length and width) of individual structures were positively correlated ($R^2$ = 0.58). Interestingly, we often found that PSDs were associated with multiple clathrin structures, similar as to what we observed with gSTED imaging (*Figure 1B and D*). We noted that two distinct populations could be observed based on morphological characteristics and distinguished between the primary and secondary clathrin structure based on size. The largest was classified as the primary structure and we found that this structure most likely corresponds to the EZ, as these were also the most closely linked to the PSD (*Figure 3—figure supplement 1A, B*). The secondary, smaller structures were between 50 and 100 nm in diameter (*Figure 3—figure supplement 1C*), similar to the reported size of endocytic vesicles (*Kirchhausen and Harrison, 1981*; *Pearse and Crowther, 1987*). These smaller structures also appeared more circular and further away from the PSD (*Figure 3—figure supplement 1B, D*), further suggesting that these smaller secondary structures are endocytic vesicles that perhaps budded off from the edge of the EZ.

## Endocytic proteins are differentially retained at perisynaptic sites

Apart from clathrin, only a few other endocytic components have been suggested to be part of the EZ. Among these proteins are synaptotagmin-3 (*Awasthi et al., 2019*), PICK1 (*Fiuza et al.,*

*2017*) and CPG2 (*Cottrell et al., 2004*). Moreover, the presence of dynamin2 and AP2 at the site of clathrin-coated pits in spines suggest that these proteins could also be part of the EZ (*Rácz et al., 2004*). However, it remains unknown whether these and other endocytic proteins are stably accumulated at the EZ, or whether these are perhaps transiently recruited only during endocytic events. To begin to address this, we first determined the localization of 12 well-known endocytic proteins using confocal microscopy and live-cell imaging. Among these proteins are the well-known F-BAR and N-BAR proteins like FCHO1, syndapin-1 (Sdp1), syndapin- 2 (Sdp2) and amphiphysin (Amph), β2-adaptin, a subunit of the membrane proteins AP2, scission protein dynamin2 (Dyn2) and other adaptor proteins like Eps15, PICALM, intersectin-1 long (Itsn1L) and epsin-2 (Epsn2). In addition, we included HIP1R and CPG2 that can couple the endocytic machinery to the actin cytoskeleton (*Chen and Brodsky, 2005*; *Engqvist-Goldstein et al., 2001*; *Loebrich et al., 2016*; *Wilbur et al., 2008*). We found that most of these proteins localized at perisynaptic sites in a punctate manner, similar to clathrin (*Figure 4B*). Sdp1, Sdp2, and Amph showed a more diffuse signal within the spine and dendritic shaft. For Amph, clear puncta associated with the PSD could be detected occasionally; however, Sdp1 and Sdp2 did not seem to be enriched in distinct puncta and were not further analyzed. In this experiment, we found that 66.8% ± 6.9% of PSDs was associated with GFP-CLCa (*Figure 4A and C*). We found that the fraction of PSDs associated with HIP1R, β2-adaptin, Dyn2, CPG2, Eps15, and Itsn1L was similar to the percentage of clathrin-associated PSDs. In contrast, PICALM, Epsn2, Amph, and FCHO1 were less frequently found in association with the PSD (*Figure 4B and C*). Thus, HIP1R, β2-adaptin, Dyn2, CPG2, Eps15, and Itsn1L appear associated with the PSD and could be intrinsic components of the EZ. To validate that the localization of these proteins was not disrupted as a result of overexpression, we endogenously tagged Eps15, Itsn1 and Dyn2 using CRISPR/Cas9-mediated genome editing (*Willems et al., 2020*). We attempted to generate knockin constructs for multiple AP2 subunits but were unfortunately not able to reliably tag these subunits. Importantly, for endogenously tagged Eps15, Itns1 and Dyn2 we found a similar distribution and perisynaptic localization as we found in the exogenous expression experiments (*Figure 4D*, *Figure 4—figure supplement 1*). Similarly, endogenous labeling using antibodies for Eps15, Itsn1, and Dyn2 confirmed that these proteins are located at perisynaptic sites (*Figure 4— figure supplement 1*).

Next, to test whether these endocytic proteins were stably associated with the PSD, we performed time-lapse experiments on neurons co-expressing Homer1c and a fluorophore-tagged endocytic protein. Neurons were imaged for 10 min at 20-s time intervals. We found very distinct behaviors in the dynamics of endocytic proteins. While some proteins only transiently occurred at perisynaptic sites (e.g. FCHO1; *Figure 5A*), other proteins appeared stable over the entire duration of the acquisition (e.g. CPG2; *Figure 5B*). Consistent with our previous observations in fixed neurons (*Figure 4C*), the percentage of PSDs associated with a clear endocytic protein structure for the entire duration of the acquisition was high for HIP1R, β2-adaptin, Dyn2, CPG2, Eps15, and Itsn1L, and much lower for PICALM, FCHO1, Epsn2, and Amph (*Figure 5C*). From these live-cell acquisitions, we determined the lifetime of events where these proteins were enriched at perisynaptic sites. Interestingly, when plotted as histograms, a clear bimodal distribution of lifetimes was observed (*Figure 5D*). Endocytic proteins accumulated either briefly ( < 3 min) or appeared persistent ( > 9 min) at perisynaptic sites (*Figure 5D*). These data also indicated that HIP1R, β2-adaptin, Dyn2, CPG2, Eps15, and Itsn1L are stable components that are generally long-lived (*Figure 5D*). In contrast, the average lifetimes of FCHO1 (2.74 ± 1.7 min), PICALM (1.73 ± 0.20 min), Epsn2 (3.00 ± 0.20 min), and Amph (2.73 ± 1.7 min) at perisynaptic sites were significantly lower compared to the average lifetime of CLCa. Notably, the lifetime of these short-lived events is comparable to the duration of endocytic events (~2 min), suggesting that these proteins are transiently recruited upon the induction of endocytosis. To determine the turnover of the long-lived endocytic proteins at perisynaptic sites, we performed FRAP experiments. Except for CPG2, all endocytic proteins showed considerably higher turnover than clathrin (percentage of recovery after 10 min CPG2: 29.4% ± 3.7%, HIP1R: 59.2% ± 5.2%, β2-adaptin: 56.7% ± 3.8%, Eps15: 88.1 ± 3.8, Itsn1L: 93.9 ± 8.7) (*Figure 5E*, *Figure 5— figure supplement 1*). Taken together, these experiments reveal that apart from clathrin, HIP1R, β2-adaptin, Dyn2, CPG2, Eps15, and Itsn1L are also integral components of the perisynaptic EZ, while PICALM, FCHO1, Epsn2, and Amph only appear transiently at perisynaptic sites, perhaps to initiate or facilitate endocytosis.

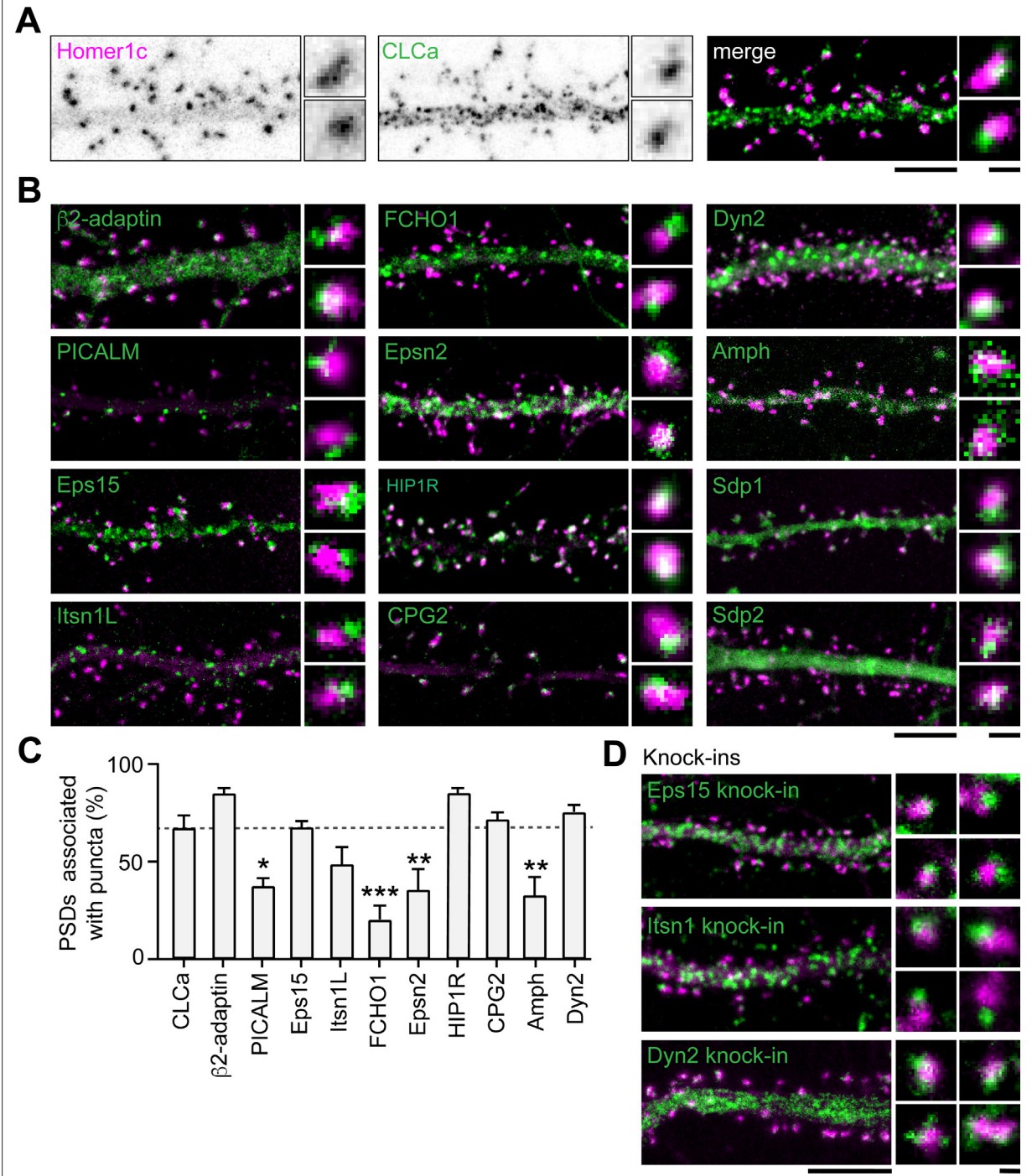

**Figure 4.** Endocytic accessory proteins are associated with the PSD. (**A**) Example images of dendrites expressing Homer1c-mCherry to mark PSDs and GFP-CLCa visualized as black and white images (left, middle panel), and merge (right panel). (**B**) Example images of neurons co-expressing tagged endocytic proteins relative to Homer1c. (**A–B**) Scale bars: 5 μm, zoom dimensions: 1 μm. (**C**) Percentage of synapses associated with endocytic proteins, represented as mean ± SEM. Relative to Homer1c-CLCa association (N = 8), PICALM-mCherry (N = 6, p < 0.05), FCHO1-mCherry (N = 6, p < 0.001), Epsn2-mCherry (N = 9, p < 0.01), and Amph-mCherry (N = 6, p < 0.01) were significantly less often associated with the PSD, while β2-adaptin- GFP (N = 6), GFP-Eps15 (N = 5), GFP-Itsn1L (N = 5), HIP1R-GFP (N = 9), GFP-CPG2 (N = 5), Dyn2-GFP (N = 5) were not different from GFP-CLCa. (**D**) Example images of neurons expressing Homer1c-ALFA labeled with Cy3 (magenta) and endogenously GFP-tagged endocytic proteins using CRISPR/Cas9-mediated genome editing (green). Scale bar: 5 μm, zoom: 500 nm. Data represented as mean ± SEM.

The online version of this article includes the following source data and figure supplement(s) for figure 4:

**Source data 1.** Excel sheet with numerical data represented as plots *Figure 4C*.

*Figure 4 continued on next page*

*Figure 4 continued*

**Figure supplement 1.** Endogenous labeling of endocytic proteins in neurons.

**Figure supplement 1—source data 1.** Excel sheet with numerical data represented as a plot in *Figure 4—figure supplement 1A*.

## Endocytic proteins have distinct spatial organization relative to the clathrin coat at the EZ

The presence of multiple endocytic adaptor proteins and their stable retention at perisynaptic sites suggests that these proteins might be an integral part of the EZ. First, we applied two-color gSTED on Halo-CLCa co-transfected with the stably retained endocytic proteins fused to GFP and stained for endogenous Homer1b/c to localize the PSD. We indeed found that Eps15, Itsn1L, Dyn2, β2-adaptin, and HIP1R all colocalize with clathrin next to the PSD (*Figure 5—figure supplement 2*).

Next, to dissect the spatial organization of endocytic proteins relative to the clathrin structure at the EZ we used two-color SMLM. Halo-CLCa was co-transfected with a GFP-tagged endocytic protein to efficiently label and acquire high-density localization maps in two channels. Strikingly, we found that β2-adaptin, Eps15, and Itsn1L were often distributed in smaller patches around and sometimes within the EZ marked by CLCa (*Figure 6A–C*). HIP1R showed a more homogenous distribution and often colocalized with the EZ entirely and even surrounding the EZ (*Figure 6D*). Dyn2 showed an overall more homogenous distribution, similar to HIP1R (*Figure 6E*). However, we also found examples where Dyn2 localized in small clusters at the edge of the EZ. To analyze these distributions quantitatively, we manually selected regions around clathrin structures in dendritic spines for further analysis. We then used DBScan to determine the outline of the EZ marked by CLCa. We first mapped the absolute distance of localizations to the border of the EZ averaged over a population of EZs and found that all the endocytic proteins analyzed here peaked within 25 nm from the edge of the clathrin structure (*Figure 6F*). However, since individual EZs can vary in size, we next mapped the density of endocytic proteins in rings that were set in size relative to the clathrin structure. For each EZ, we defined eight incremental rings that were scaled with proportion to the outline of the EZ and binned the density of localizations within each of these rings (*Figure 6G*). As expected, when plotting the relative fraction of Halo-CLCa localization within the rings, we found that the density of clathrin molecules was highest in the center (0–20% ring), gradually decreased towards the outer ring (80–100%) and dropped to close to zero in the rings surrounding the EZ (100–160% rings). In contrast, when we plotted the relative density of Homer1c localizations relative to CLCa, we found a clear separation of these distributions (*Figure 6H*), further validating the analysis. When analyzing the endocytic adaptor proteins relative to the EZ, we again found that the relative density of Eps15, β2-adaptin, and Itsn1L peaked within the EZ, but close to the edge of the EZ (*Figure 6I*). The profiles of HIP1R and Dyn2 localizations (*Figure 6G*) showed less clear peaks, indicating a more homogenous distribution of these proteins within the EZ.

## Interactions with the PSD, but not with the membrane or actin cytoskeleton are required for the perisynaptic localization of the EZ

The differential dynamics and nanoscale organization of endocytic proteins at perisynaptic clathrin structures suggests that the EZ is a highly organized structure where several endocytic proteins are assembled. The mechanisms that retain the EZ at this particular position, however, are not fully understood. The EZ is coupled to the PSD via Shank-Homer-Dyn3 interactions (*Lu et al., 2007*; *Petrini et al., 2009*; *Scheefhals et al., 2019*). In addition, we now identified several new EZ components that can couple to the plasma membrane, for example via AP2, or the actin cytoskeleton, for example via CPG2 and HIP1R (*Loebrich et al., 2016*; *Saffarian et al., 2009*), suggesting that these modes of interaction could also contribute to the positioning of the EZ. To test this, we interfered with several of these connections. First, we tested whether Shank knockdown (KD), which we showed previously uncouples the EZ from the PSD (*Scheefhals et al., 2019*), also leads to the loss of these newly identified EZ components (*Figure 7A and B*). Indeed, we found that Shank-KD did not only reduce the number of clathrin-positive PSDs as found before (0.5 ± 0.1 relative to control), but also reduced the association of the PSD with other endocytic proteins (*Figure 7B*). We found that Shank-KD significantly reduced Homer1c-Eps15 (0.63 ± 0.1), Homer1c-Itsn1L (0.6 ± 0.1) and Homer1c-β2-adaptin (0.55 ± 0.1) coupling compared to control (*Figure 7B*). Interestingly, HIP1R (0.89 ± 0.04), Dyn2 (0.91 ± 0.1), and CPG2 (0.86 ± 0.1) were not uncoupled from the PSD (*Figure 7B*). Together, these findings

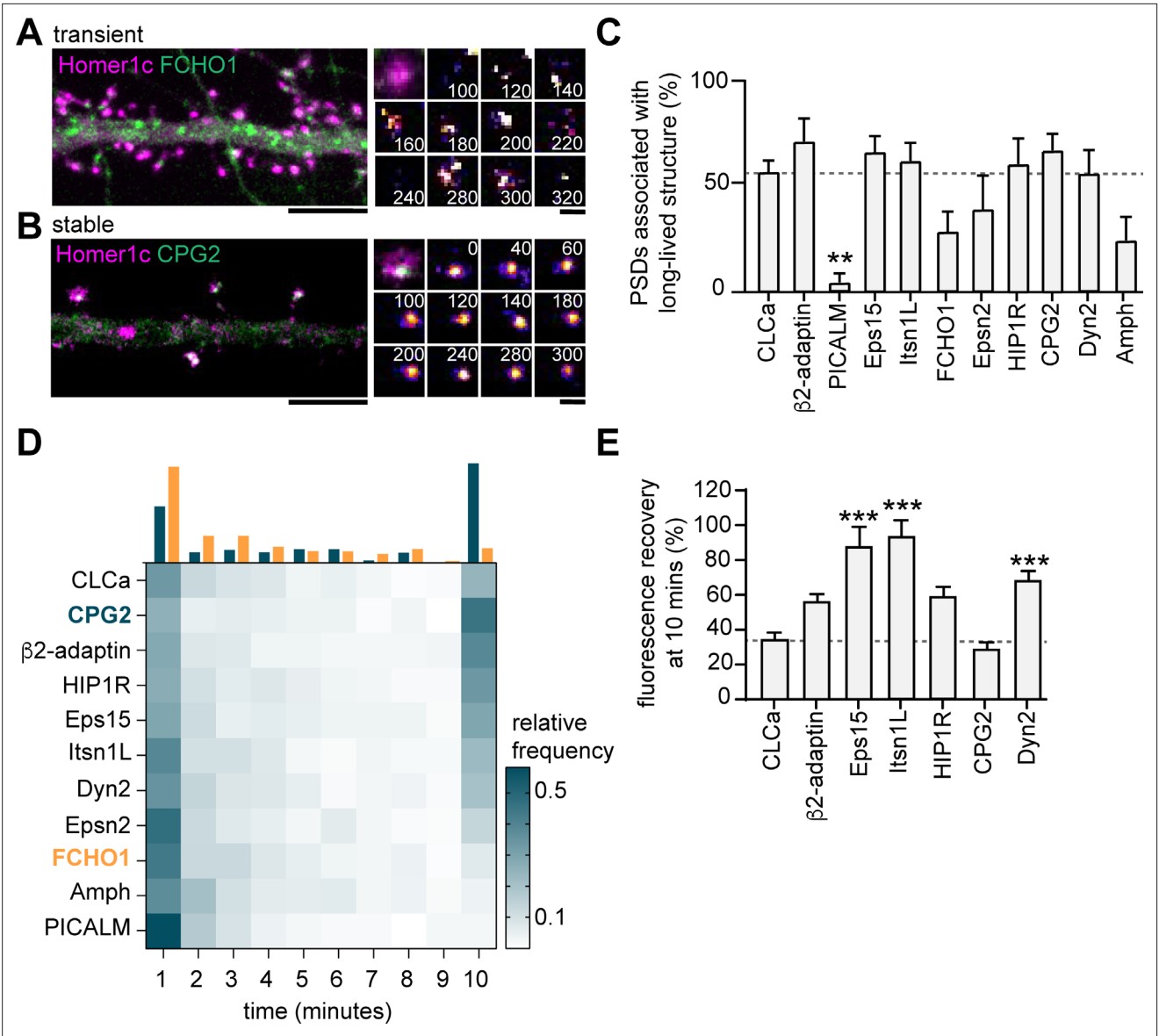

**Figure 5.** Endocytic proteins are differentially associated with the PSD. (**A**) Example image of FCHO1 (green) that is transiently associated with Homer1c (magenta). Zooms show temporal recruitment of FCHO1. Scale bar: 5 µm, zoom: 500 nm. (**B**) Example image of CPG2 that is stably associated with Homer1c. Zooms show temporal dynamics of CPG2. Scale bar: 5 µm, zoom: 500 nm. (**C**) Percentage of synapses that contain at least one stable structure (persisting for >9 min). Only PICALM-mCherry (N = 5, p < 0.01) was significantly less often stably associated with the PSD compared to GFP-CLCa (N = 6). β2-adaptin-GFP (N = 6), GFP-Eps15 (N = 6), GFP-Itsn1L (N = 6), FCHO1-mCherry (N = 5), Epsn2-mCherry (N = 5), HIP1R-GFP (N = 6), GFP-CPG2 (N = 8), Amph-mCherry (N = 5), Dyn2-GFP (N = 7), were not different from GFP-CLCa. (**D**) Heatmap visualizing the frequency distribution of the lifetime of endocytic proteins associated with the PSD. The histogram on top is an example of FCHO1 (orange) that is mostly short-lived, and CPG2 (blue) that is mostly stable, plotted as relative frequency. (**E**) Summary graph of the recovery 10 min after FRAP for GFP-Eps15 (n = 23, p < 0.001), GFP-Itsn1L (n = 20, p < 0.001), HIP1R-GFP (n = 44, p < 0.01), Dyn2-GFP (n = 51, p < 0.001) had significantly higher turnover compared to GFP-CLCa (n = 32). GFP-CPG2 (n = 22) and β2-adaptin GFP (n = 13) were not different compared to GFP-CLCa. Data plotted as mean ± SEM.

The online version of this article includes the following source data and figure supplement(s) for figure 5:

**Source data 1.** Excel sheet with numerical data represented as plots in *Figure 5C, D and E*.

**Figure supplement 1.** FRAP curves of endocytic proteins compared to CLCa.

**Figure supplement 1—source data 1.** Excel sheet with numerical data represented as plots in *Figure 5—figure supplement 1A,B,C,D,E,F*.

**Figure supplement 2.** Endocytic proteins colocalize with the EZ.

**Figure supplement 2—source data 1.** Excel sheet with numerical data represented as plots in *Figure 5—figure supplement 2B,C*.

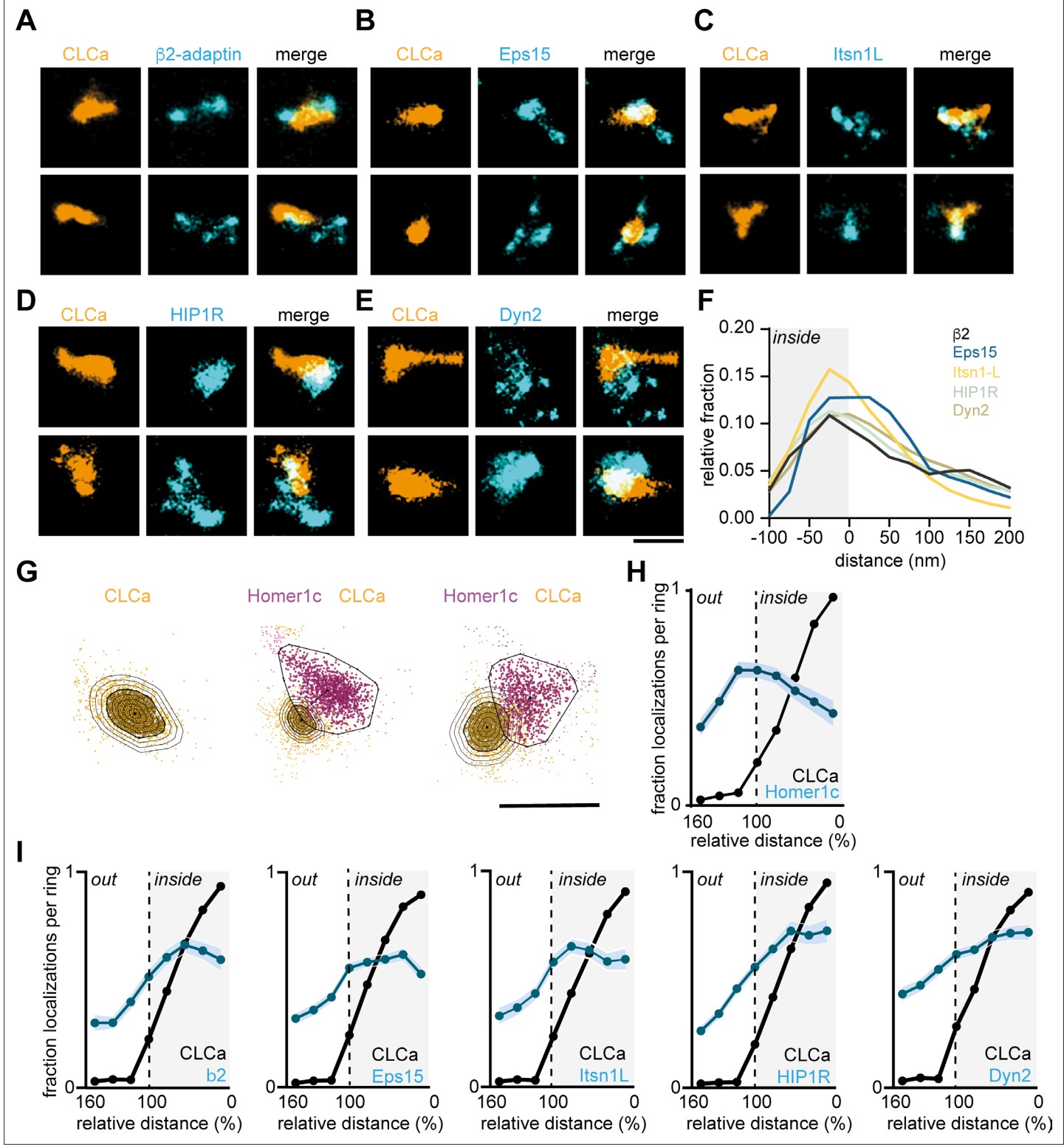

**Figure 6.** Endocytic proteins have distinct spatial organization relative to the clathrin structure marking the EZ. (**A–E**) High-resolution example images of Halo-CLCa labelled with JF647 (orange) co-expressed with endocytic proteins fused to GFP labelled with CF568 (cyan). Scale bar: 500 nm. (**F**) Histogram visualizing the relative frequency of the distance of individual localizations relative to the border of CLCa. (**G**) Example plots of CLCa with rings based on the size of the structure (left panel), and example plots of individual CLCa and Homer1c localizations. The border outlining the EZ (bold black outline around the orange localizations) represents the 100% mark in graphs H and I. Scale bar: 500 nm. (**H**) Fraction of localizations per ring. Dotted line depicts the border of CLCa (Homer1c n = 65, CLCa n = 66). (**I**) Fraction of localizations per ring. Dotted line depicts the border of CLCa. β2-adaptin (n =

*Figure 6 continued on next page*

*Figure 6 continued*

87), GFP-Eps15 (n = 126), GFP-Itsn1L (n = 58), GFP-HIP1R (n = 72), GFP-Dyn2 (n = 82).

The online version of this article includes the following source data for figure 6:

**Source data 1.** Excel sheet with numerical data represented as plots *Figure 6F, H1*.

show that PSD-EZ coupling via Shank proteins is necessary for the integrity of the EZ. Next, to test if other mechanisms contribute to EZ maintenance we tested whether alterations in actin dynamics, membrane binding capacity or depleting specific EZ components would lead to a reduction of EZs (*Figure 7C*). Interestingly, we found that disrupting the integrity of the actin cytoskeleton did not result in a clear reduction of the PSD-EZ association. The actin depolymerization drug Latrunculin B slightly decreased the PSD-EZ association (0.81 ± 0.1), but this was not statistically significant. The Arp2/3 inhibitor CK-666 (0.98 ± 0.1) also did not significantly alter PSD-EZ association (*Figure 7D and F*), suggesting that disrupting actin dynamics does not disrupt positioning of the EZ. Jasplakinolide, an F-actin stabilizing drug, resulted in significantly more PSD-EZ association compared to control (1.27 ± 0.04, p < 0.05). To test specific EZ-actin interactions, we assessed whether the binding of clathrin to HIP1R is necessary for EZ maintenance. Overexpression of a clathrin-light chain mutant that is unable to bind HIP1R (GFP-CLCb-EED/QQN) (*Chen and Brodsky, 2005*; *Poupon et al., 2008*), did not affect the localization of the EZ (*Figure 7E and F*), further indicating that coupling to the actin cytoskeleton is not a primary mechanism for maintaining the EZ. In addition, we considered CPG2 as a candidate protein as it has been shown to couple the actin cytoskeleton to the membrane (*Loebrich et al., 2016*; *Loebrich et al., 2013*) and we found it is an exceptionally stable EZ protein. However, knockdown of endogenous CPG2 using established shRNAs (*Cottrell et al., 2004*) did not affect PSD-EZ coupling (*Figure 7E*). The third mechanism that could allow the perisynaptic localization of the EZ involves interactions with the plasma membrane. To address this, we overexpressed a mutant form of the AP2 mu2 subunit that is unable to interact with PIP2 (AP2m2-P1) and was shown to hamper receptor inter-nalization (*Raman et al., 2014*), but we found no change in the fraction of EZ-positive PSDs (1.1 ± 0.1), suggesting that coupling to the membrane via AP2-PIP2 interactions does not affect EZ maintenance (*Figure 7E and F*). Lastly, we checked if removing specific, stable endocytic components would affect EZ positioning. Itsn1L is multi-domain scaffold protein that interacts with several endocytic proteins to orchestrate endocytosis and could thus have a central role as scaffold in the EZ. However, Itsn1L knockdown did not alter the fraction of EZ-positive PSDs (*Figure 7E and F*). Altogether, based on these mechanistic experiments, we conclude that the EZ is assembled from a distinct set of endocytic proteins and is maintained and positioned primarily by interactions with the PSD.

## Plasticity-induced reorganization of the EZ

The EZ is remarkably stable under basal conditions, but whether this structure is reorganized during long-term synaptic plasticity remains relatively unexplored. To test this, we first compared two distinct paradigms to induce LTD: one triggered by NMDAR activation (NMDAR-LTD) (*Lee et al., 1998*) and one induced by activation of group I mGluRs (mGluR-LTD) (*Huber et al., 2001*). We found that induction of mGluR-LTD by a 5 minute-application of DHPG, significantly increased the number of PSDs associated with multiple CCSs (control: 0.95 ± 0.1, 5' DHPG: 2.1 ± 0.4). Strikingly, the effect of DHPG was transient and the number of PSDs associated with multiple CCSs returned to control levels after 10 min (*Figure 8A and B*). This effect was blocked by the selective mGluR5 antagonist MPEP, indicating that this effect was mediated by mGluR5 activation. Intriguingly, application of NMDA had only a small, non-significant effect (control: 1.03 ± 0.08, 5' NMDA: 1.47 ± 0.19), indicating that these protocols differentially affect the perisynaptic CCSs. Next, to see if synapse-potentiating paradigms induce EZ reorganization, we induced chemical LTP (cLTP) using glycine and bicuculin for 5 min (*Lu et al., 2001*; *Figure 8D*). We found that cLTP resulted in an increase in the number of PSDs containing multiple CCSs 30 min after induction (control: 17.6% ± 2.8%, 30' cLTP: 29.1% ± 2.7%, p < 0.01). Remarkably, in contrast to cLTD this effect was long-lasting, up to 2 hr after cLTP induction (*Figure 8E*). The effect was blocked in the presence of AP5, an NMDAR antagonist, indicating that the observed effect is NMDAR-mediated (*Figure 8F*). Next, we performed live-cell gSTED to observe CCSs over time within individual synapses after cLTP. Indeed, also in these live-cell experiments we observed that the EZ is structurally altered after cLTP induction (*Figure 8G*). In some cases, multiple CCSs could be

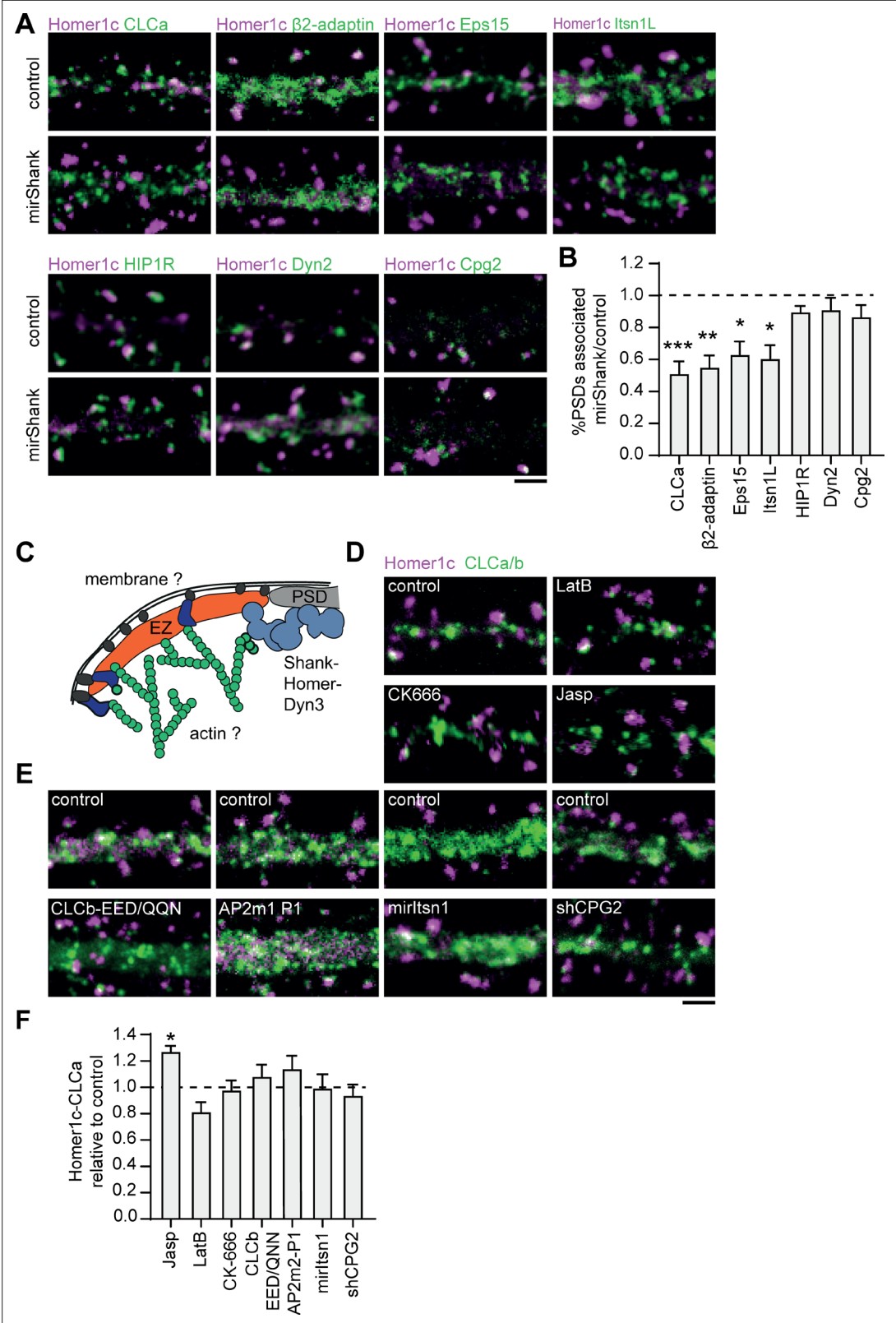

**Figure 7.** Interactions with the PSD, but not with the membrane or actin cytoskeleton are required for positioning of the EZ. (**A**) Example images of dendrites expressing Homer1c-ALFA and endocytic proteins fused to GFP co-expressed with control or mirShank-mCherry construct. Scale bar: 2 μm. (**B**) Fraction of PSDs associated with an EZ after Shank-KD relative to control plotted as mean ± SEM. GFP-CLCa (N = 8, p < 0.001), β2-adaptin (N = 10, p < 0.01), Eps15 (N = 12, p < 0.05), Itsn1L (N = 10, p < 0.05), HIP1R (N = 11 p > 0.05), Dyn2 (N = 12, p > 0.05), CPG2 (N = 13, p > 0.05). (**C**) Illustration of

*Figure 7 continued on next page*

eLife Research article

Cell Biology | Neuroscience

possible mechanisms that could maintain the EZ adjacent to the PSD. (D) Example images of dendrites co-expressing Homer1c-ALFA and GFP-CLCa in dendrites treated with LatB, CK666, or Jasp. Scale bar: 2 µm. (E) Example images of dendrites expressing control constructs, CLCb-EED/QQN (left panel) and AP2m2-P1 (middle panel), or Itsn1 KD construct (mirItsn1; right panel). Scale bar: 2 µm. (F) Fraction of EZ-associated PSDs relative to control, plotted as mean ± SEM. Jasp (N = 4), LatB (N = 8), CK666 (N = 12), CLCb-EED/QQN (N = 7), AP2m2-P1 (N = 5), mirItsn1 (N = 11), shCPG2 (N = 8).

The online version of this article includes the following source data for figure 7:

**Source data 1.** Excel sheet with numerical data represented as plots *Figure 7D and F*.

observed already 5 min after cLTP induction, however these structures rapidly disappeared, perhaps indicating transient clathrin coat formation. Other examples clearly showed reorganization of the EZ after 5 min cLTP that resulted in multiple CCSs after 30 min. Interestingly, we noted that the appearance of new CCSs was preferentially observed at PSDs that increased in size in response to the cLTP protocol, while PSDs that were not associated with multiple CCS after 30 min remained the same size (*Figure 8H*). These results thus suggest that long-term reorganization of the EZ preferentially occurs at PSDs that undergo activity-dependent remodeling. We found that CLCa area was not different between control and cLTP (control: 0.06 ± 0.007, cLTP: 0.05 ± 0.009) (*Figure 8I*). Similarly, live-cell confocal imaging showed that CLCa intensity was not affected after cLTP (*Figure 8J*), indicating that there is no recruitment of clathrin to spines, but suggests that the EZ undergoes remodeling, or perhaps splits in response to synaptic activity. These data show that synaptic activity patterns that induce long-term alterations in synaptic strength are accompanied by structural reorganization of the perisynaptic endocytic machinery at excitatory synapses.

## Discussion

Localized endocytosis of synaptic receptors at the EZ is essential for the maintenance and activity-directed changes in the composition of the synaptic membrane. However, the molecular composition and organization of the EZ has remained largely elusive. Here, we present evidence that the EZ is a highly unique clathrin structure that contains a defined arsenal of endocytic proteins, is differentially retained at the EZ and highly organized at the nanoscale level with respect to the clathrin assembly. Moreover, we show that the EZ is reorganized in response to synaptic activity.

While CCSs in the dendritic shaft form a highly heterogenous population, including small, fast-moving particles, as well as larger stationary structures, the EZ is remarkably stable with little variation from spine to spine. This heterogeneity in the dendritic shaft likely resembles CCSs found in other cell types, where small clathrin structures represent transient endocytic pits or intracellular vesicles while the large patches are stable, membrane-attached structures (*Grove et al., 2014*; *Leyton-Puig et al., 2017*; *Saffarian et al., 2009*). Our findings on the EZ are in line with previous studies, showing little exchange of clathrin at the EZ, and highly similar morphology and dynamics from spine to spine (*Blanpied et al., 2002*; *Petrini et al., 2009*; *Rosendale et al., 2017*; *Scheefhals et al., 2019*). Together, these results indicate that based on the morphological and dynamic behavior of clathrin, the EZ can be distinguished from other clathrin assemblies found in the dendritic shaft.

Importantly, our findings significantly expand on the notion that the EZ is a perisynaptic site of endocytosis by identifying several key endocytic proteins that reside at the EZ. Based on our live-cell imaging, quantitative super-resolution imaging and mechanistic studies, we conclude that the early-phase endocytic proteins β2-adaptin, Eps15, and Itsn1L are stable EZ residents that localize preferentially at the edge of the EZ, which is surprisingly similar to findings on flat clathrin lattices in non-neuronal cells (*Sochacki et al., 2017*). The accumulation of the AP-2 complex and its binding partner Eps15 at the periphery of the EZ, likely contributes to the efficient capture of cargoes, that is synaptic membrane proteins, and their local uptake via endocytosis. In addition, we recently reported that Itsn1, a multi-domain scaffold protein that coordinates different aspects of endocytosis (*Pechstein et al., 2010*; *Evergren et al., 2007*; *Hussain et al., 2001*), also facilitates mGluR-mediated AMPAR trafficking (*van Gelder et al., 2020*). Furthermore, removing Shank proteins, which was previously shown to uncouple the EZ from the PSD (*Lu et al., 2007*; *Scheefhals et al., 2019*), specifically uncoupled the early-phase proteins, indicating that these proteins are indeed stable residents and coupled to the EZ.

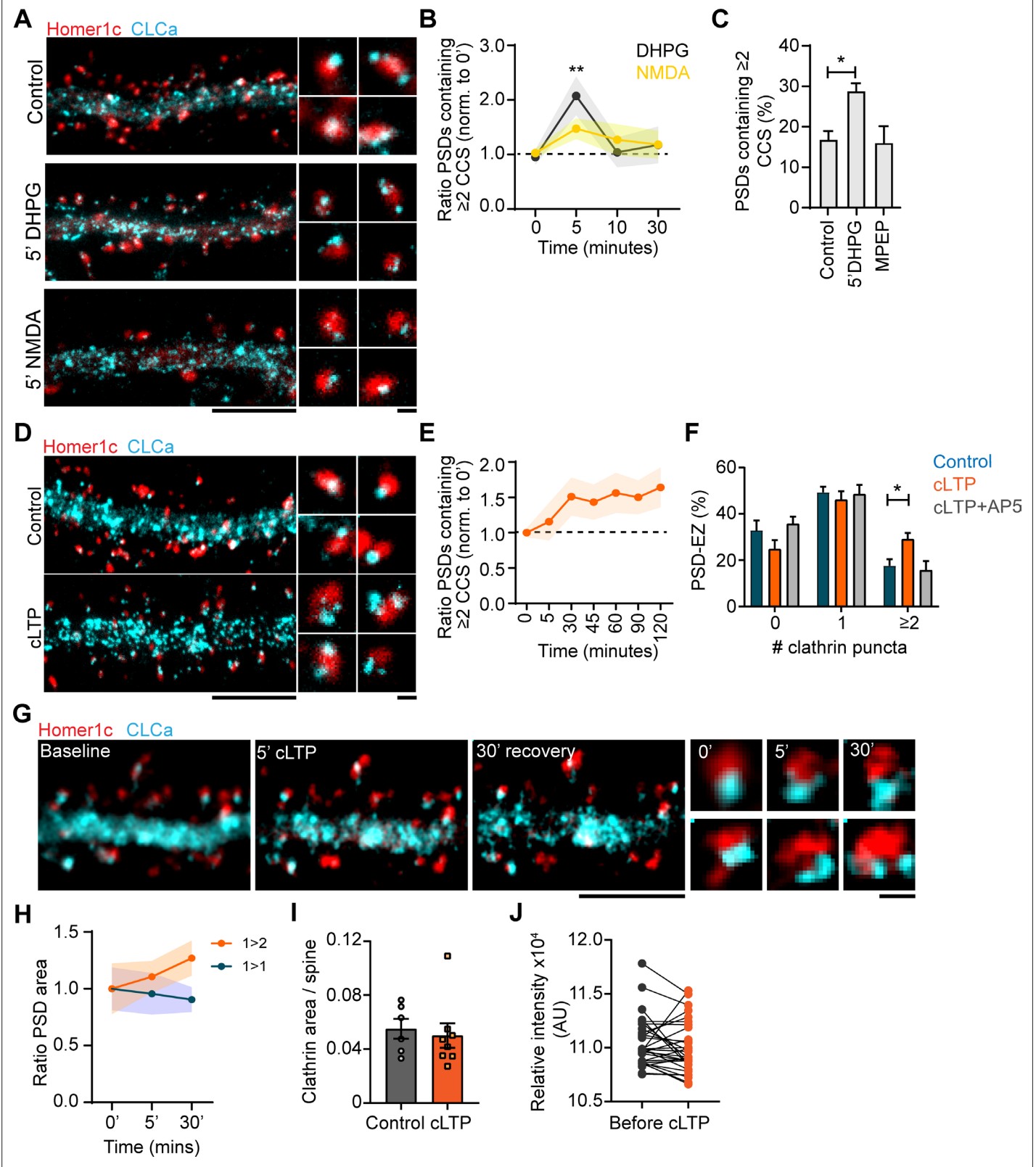

**Figure 8.** Plasticity-induced reorganization of the EZ. (**A**) Example gSTED images of Homer1c-ALFA labeled with Cy3, and CLCa-GFP labeled with At647N under basal conditions (upper panel, control), after 5 min of DHPG (100 μM, middle panel) and after 5 min of NMDA (20 μM, lower panel). Scale bar: 5 μm, zoom: 500 nm. (**B**) Quantification of the number of PSDs associated with multiple CCSs after different timepoints of DHPG and NMDA bath application, normalized to timepoint 0. DHPG significantly increased the number of PSDs associated with multiple CCSs after 5 min (timepoint 0: N = 19,

*Figure 8 continued on next page*

*Figure 8 continued*

timepoint 5: N = 11, p > 0.01), while NMDA did not alter PSD-EZ association (timepoint 0: N = 13, timepoint 5: N = 7, p > 0.05). (**C**) Percentage of PSDs associated with multiple CCSs. Five min DHPG significantly increases PSD-EZ association (Control: N = 17, DHPG: N = 13, p < 0.05). MPEP blocked the effect of DHPG (N = 10, p > 0.05). (**D**) Example gSTED images as described in A, under basal conditions (upper panel) and after cLTP (lower panel). Scale bar: 5 µm, zoom 500 nm. (**E**) Quantification of the number of PSDs associated with multiple CCSs after cLTP, normalized to timepoint 0 (N = 8–13). (**F**) Number of CCSs per PSD under basal conditions and after 5 min cLTP followed by 25 min recovery. At timepoint 30, the number of CCSs per PSD was significantly increased (Control: N = 9, cLTP: N = 11, p < 0.05) and the effect was completely blocked in the presence of AP5 (N = 8). (**G**) Example images of live-cell gSTED on Homer1c-mCherry and Halo-CLCa labeled with JF646. Three sequential images were taken before cLTP (left), directly after (middle left) and after recovery (middle right). Two example spines were selected where reorganization of CLCa was observed (right). Scale bar: 5 µm, zoom: 500 nm. (**H**) Quantification of a subset of PSDs that increased in size (orange), or remained the same size during imaging (blue). PSDs that were enlarged in response to cLTP were associated with multiple CCSs (N = 4). (**I**) GFP-CLCa area per spine under basal conditions and after cLTP (Basal: N = 6, cLTP: N = 8). (**J**) Relative intensity of GFP-CLCa before and after cLTP imaged using live-cell confocal imaging (n = 31).

The online version of this article includes the following source data for figure 8:

**Source data 1.** Excel sheet with numerical data represented as plots *Figure 8B, C, E, F, H1 and J*.

The later-phase proteins HIP1R, Dyn2, and CPG2 were also stably associated with the EZ, but this association seemed independent of PSD-EZ coupling via Shank. Interestingly, both Dyn2 and HIP1R were more widely distributed. For Dyn2 we found that in most cases, the distribution of Dyn2 at the EZ was diffuse, but the clear differential localization patterns between different EZs may indicate that Dyn2 relocates to the EZ upon endocytosis as has been suggested before (*Rosendale et al., 2017*). Unfortunately for CPG2, we were not able to obtain high-density localization maps and were unable to draw any conclusions about the nanoscale organization of CPG2 at the EZ. Taken together, it is likely that the later-phase proteins are not directly coupled to the EZ under basal conditions but rather preferentially reside in close proximity via other interactions. Finally, we found that PICALM, FCHO1, Epsn2, and Amph were associated with the EZ to a much lesser extent and appeared only transiently at perisynaptic sites. Interestingly, Amph, Epsn2, and FCHO1 are BAR proteins, and preferentially bind curved membranes. Thus, perhaps these proteins are only transiently recruited to the EZ upon induction of endocytosis and the associated increase in membrane curvature.

We found that expression of a dominant-negative form of AP2-mu2 that interferes with the clathrin-PIP2 binding did not abrogate EZ positioning, suggesting that AP2 is not involved in EZ coupling to the membrane or membrane anchoring is not a prerequisite for EZ positioning. Similarly, we found that abrogating EZ-actin cytoskeletal interactions, did not disrupt the localization of the EZ at perisynaptic sites. Both HIP1R and CPG2 are actin-binding proteins that couple essential components of the endocytic machinery to the actin cytoskeleton, and were previously shown to facilitate endocytosis (*Chen and Brodsky, 2005*; *Cottrell et al., 2004*; *Engqvist-Goldstein et al., 2001*; *Loebrich et al., 2016*; *Loebrich et al., 2013*; *Nedivi, 1999*; *Wilbur et al., 2008*). Moreover, HIP1R has been shown to be involved in maintaining stable clathrin structures (*Grove et al., 2014*; *Saffarian et al., 2009*). However, removing CPG2 or interfering with the HIP1R-Clathrin interaction did not disrupt the localization of the EZ, suggesting that although CPG2 and HIP1R are stable EZ residents they are not involved in maintaining the EZ. Moreover, disruption of the actin cytoskeleton with did not abrogate PSD-EZ association. Thus, the actin cytoskeleton does not seem to have a prime structural role in maintaining or positioning the EZ.

We found that mGluR-LTD transiently increased the number of CCSs per PSD, while NMDAR-mediated LTD did not. Interestingly, while both NMDAR-LTD and mGluR-LTD lead to a reduction of surface AMPARs, activation of NMDARs and mGluRs trigger different signaling pathways and internalize distinct pools of AMPARs (*Casimiro et al., 2011*; *Oliet et al., 1997*; *Waung et al., 2008*). It would thus be of interest to further delineate how these receptors differentially act on EZ structure and function. Strikingly, we found that cLTP, which is typically associated with increased exocytosis of AMPARs, induced long-lasting reorganization of clathrin at the EZ. Nevertheless, a recent computational study suggested that also LTP is associated with increased endocytic flux (*Sumi and Harada, 2020*). Similarly, an EM study showed that cLTP was associated with the appearance of multiple clathrin-coated pits per PSD (*Puchkov et al., 2011*). Moreover, it has been shown that LTP-induced internalization of SK channels contributes to synaptic potentiation (*Lin et al., 2008*). Thus, perhaps the reorganization of perisynaptic CCSs facilitates changes in endocytic rate and modulates LTP induction and maintenance. On longer time scales, the reorganization of spine CCSs might endow potentiated

spines with the ability to more readily recycle synaptic receptors and thus respond more efficiently to changes in synaptic activity.

Taken all together, we found that the EZ is a highly organized clathrin structure where endocytic proteins are differentially retained and stabilized. This distinct organization likely facilitates the efficient capture and endocytosis of synaptic membrane proteins close to the PSD. Moreover, for the first time, we show that the EZ is reorganized in response to synaptic activity. These findings motivate further investigation into the molecular composition, the mechanisms that control the recruitment and activation of individual EZ components and the coupling of the EZ to the intracellular endosomal system. Elucidating these aspects of the EZ will contribute to a better understanding of this subcellular structure in neurons that is so critical for the maintenance and activity-dependent modulation of neuronal synapses.

# Materials and methods

## Animals

All animal experiments were performed in compliance with the guidelines for the welfare of experimental animals issued by the Government of the Netherlands (Wet op de Dierproeven, 1996) and European regulations (Guideline 86/609/EEC). All animal experiments were approved by the Dutch Animal Experiments Review Committee (Dier Experimenten Commissie; DEC), performed in line with the institutional guidelines of Utrecht University.

## Primary hippocampal cultures and transfection

Hippocampal cultures were prepared from brain of embryonic day 18 (E18) Wistar rats (both genders) as described before (*Scheefhals et al., 2019*). Dissociated hippocampal neurons were plated on coverslips coated with poly-L-lysine (37.5 µg/ml, Sigma-Aldrich) and laminin (1.25 µg/ml, Roche Diagnostics) at a density of 100,000 neurons per well of a 12-well plate. Cultures were allowed to settle in Neurobasal medium (NB) supplemented with 2% B27 (GIBCO), 0.5 mM glutamine (GIBCO), 15.6 mM glutamate (Sigma-Aldrich), and 1% penicillin/streptomycin at 37 °C in 5% CO2. After 24 hr halve of the NB medium was refreshed with BrainPhys medium (BP) supplemented with SM1 supplement (Stemcell Technologies) and 1% penicillin/streptomycin, and kept at 37 °C in 5% $CO_2$. Refreshment were done weekly replacing halve of the medium with fresh supplemented BP medium. At DIV11-16 neurons were transfected with indicated constructs using Lipofectamine 2000 (Invitrogen). Before transfection, 300 µl conditioned medium was transferred to a new culture plate. For each well, 1.8 µg DNA was mixed with 3.3 µl Lipofectamine 2000 in 200 µl BP, incubated for 30 min at room temperature and added to the neurons. After 1–1.5 hr, neurons were briefly washed with BP and transferred to the new culture plate with conditioned medium with an additional 500 µl supplemented BP and kept at 37 °C in 5% $CO_2$ for 4–6 days. For the knock-ins transfection was performed at DIV3.

## DNA constructs

GFP-CLCa was a gift from Dr. Blanpied. Halo-CLCa was obtained by replacing the GFP from GFP-CLCa for a Halo-tag using Gibson assembly (NEBbuilder HiFi DNA assembly cloning kit). GFP-CPG2 was obtained by replacing the HA-tag in the HA-CPG2 construct (gift from Dr. Nedivi) using Gibson assembly. GFP-Intersectin Long (Addgene plasmid # 47395) and GFP-CLCb (EED/QQN) (Addgene plasmid # 47422) were a gift from Peter McPherson. FCHO1-pmCherryC1 (Addgene plasmid # 27690), Epsin2-pmCherryC1 (Addgene plasmid # 27673), CALM-pmCherryN1 (Addgene plasmid # 27691), Amph1-pmCherryN1 (Addgene plasmid # 27692), and Syndapin2-pmCherryC1 (Addgene plasmid # 27681) were a gift from Christien Merrifield. FKBP-β2-adaptin-GFP (*Wood et al., 2017*) and HIP1R-GFP-FKBP (Addgene plasmid # 100752) were a gift from Stephen Royle. The AP2-mu2 patch one mutant (AP2m2-P1) containing three point mutations (K341E/K343E/K345E) was a gift from Dr. Richmond (*Raman et al., 2014*). GFP-Syndapin I was a gift from Dr. Robinson. GFP-Eps15 was a gift from Dr. Van Bergen en Henegouwen. shCPG2 was a gift from Dr. Nedivi. The following constructs have been described before: Homer1c-mCherry, Homer1c-GFP, Dynamin2-GFP (*Scheefhals et al., 2019*), pSM155-mirltsn-GFP (*van Gelder et al., 2020*), GFP-CLCa knock-in construct (*Willems et al., 2020*). Homer1c-ALFA construct was cloned by replacing mCherry in Homer1c-mCherry for the ALFA tag (*Götzke et al., 2019*) and the CMV promotor was replaced by a CaMKII promotor using Gibson

assembly. The knock-in constructs for GFP-Eps15, GFP-Itsn1 and GFP-Dyn2 were designed and cloned as described previously (*Willems et al., 2020*).

## Immunocytochemistry and HaloTag labeling

Neurons were fixed between DIV16-21 with 4% paraformaldehyde (PFA, EM grade) diluted in PEM buffer (80 mM PIPES, 5 mM EGTA, 2 mM MgCl2, pH 7.4) for 10 min at 37 °C and washed three times with PBS supplemented with 100 mM glycine (PBS-gly). Then, neurons were permeabilized and blocked with 10% normal goat serum (NGS) and 0.01% Triton X-100 (TX) in PBS-gly for 30 min at 37 °C. For STED imaging, GFP and mCherry containing constructs were enhanced with corresponding rabbit anti-GFP (1:2000, MBL International Cat# 598, RRID: AB_591819) and mouse anti-mCherry (1:1000, Takara Bio Cat# 632543, RRID: AB_2307319) antibodies diluted in PBS-gly supplemented with 5% NGS and 0.01% TX, for an overnight at 4 °C. The next day, coverslips were washed three times in PBS-gly and anti-GFP was further labeled with ATTO647N-conjugated secondary antibodies (1:500, Sigma-Aldrich) and anti-mCherry was labeled with CF568-conjugated secondary antibodies (1:500, Sigma-Aldrich) for 2 hr at room temperature (RT), washed and mounted in Mowiol (Sigma-Aldrich). For SMLM on Homer-mCherry and GFP-CLCa the same procedure was used as described above, but anti-GFP was labeled with Alexa-647-conjugated secondary antibodies (Life Technologies). After 2 hr, coverslips were washed three times and kept in PBS until further use. For SMLM on Halo-CLCa combined with various endocytic proteins fused to GFP, we first performed live-labeling with Halo-JF646 (1:1000, Promega) for 15 min at RT. To label endocytic proteins, GFP was labeled with a mouse anti-GFP (1:1000, Thermo Fisher, RRID: AB_221568), and labeled with a corresponding CF568-conjugated secondary antibody (Sigma-Aldrich). Although the localization density obtained for Halo-CLCa labelled with JF646 was lower compared to GFP-CLCa labeled with primary and secondary antibodies, no difference in CLCa morphology was observed (*Figure 4—figure supplement 1*). For the knockins, GFP tagged proteins were enhanced using the polyclonal anti-GFP antibody described above and further labelled with Alexa488-conjugated secondary antibodies (1:500, Life Technologies). During the incubation of the secondary antibody Homer1c-ALFA was labeled with Cy3-conjugated FluoTAG X4 anti-ALFA (1:500 Fluotag X4, Nanotag). For the endogenous antibody labeling, the same protocol was used as described above, using anti-Eps15 (1:400, Cell Signaling Technology Cat# 8855, RRID: AB_10949158), anti-Itsn1 (1:400, Abcam Cat# ab118262, RRID: AB_10899433), and anti-Dyn2 (1:400, BD Biosciences Cat# 610263, RRID: AB_397659), further labeled with Alexa488-conjugated secondary antibodies.

## Pharmacology

For all the following experiments DIV15-16 neurons were used. Latrunculin B (20 µM, Bioconnect), jasplakinolide (20 µM, Tocris) and CK-666 (400 µM, Sigma), were incubated in prewarmed extracellular imaging buffer for 30 min at 37 °C. cLTD was induced using either NMDA (20 µM) or DHPG (100 µM) diluted in extracellular imaging buffer. NMDA was applied for 5 min. Neurons were either fixed directly after 5 min or were allowed to recover in extracellular imaging buffer and fixed at indicated timepoints. DHPG was added for 30 min unless indicated shorter. To induce cLTP, we used extracellular imaging buffer without magnesium, supplemented with freshly prepared glycine (Sigma, 300 µM) and bicuculin (25 µM, Tocris). Neurons were stimulated for 5 min after which the supplemented buffer was removed and replace with extracellular imaging buffer with magnesium.

## Confocal imaging

Confocal images were acquired with a Zeiss LSM 700 confocal laser-scanning microscope using a Plan-Apochromat 63 x NA 1.40 oil objective. Images consist of a z-stack of 5–9 planes at 0.37 µm interval, and maximum intensity projections were generated in Fiji (*Schindelin et al., 2012*) for analysis and display.

## STED imaging

Gated STED (gSTED) images were taken with the Leica TCS SP83x microscope using a HC PL APO 100 x/NA 1.4 oil immersion STED WHITE objective. The 488 nm pulsed white laser (80 MHz) was used to excite Alexa-488, 561 nm to excite CF568, and the 647 nm to excite JF646 and ATTO647N labeled proteins. JF646 and ATTO647N were depleted with the 775 nm pulsed depletion laser, and

for depleting CF568 the 660 nm pulsed depletion laser was used. The internal Leica HyD hybrid detector was set at time gate between 0.3 and 6 ns. Images were taken with a pixel size lower than 40 nm, and Z-stacks were acquired. Maximum intensity projections were generated in Fiji (*Schindelin et al., 2012*) for analysis and display.

## Live-cell imaging

Live-cell imaging was performed on a spinning disk confocal system (CSU-X1-A1; Yokogawa) mounted on a Nikon Eclipse Ti microscope (Nikon) with Plan Apo VC 100 × 1.40 NA with excitation from Cobolt Calypso (491 nm), and Jive (561 nm) lasers, and emission filters (Chroma). The microscope was equipped with a motorized XYZ stage (ASI; MS-2000), Perfect Focus System (Nikon), Evolve 512 EM-CCD camera (Photometrics), and was controlled by MetaMorph 7.7.6 software (Molecular Devices). Neurons were maintained in a closed incubation chamber (Tokai hit: INUBG2E-ZILCS) at 37 °C in extracellular imaging buffer. For high-frequency live-cell imaging (*Figure 2A–D*) images of GFP-CLCa were taken every 5 s for 5 min. For long-term live-cell imaging of Homer1c and CLCa (*Figure 2E*), images were taken every 30 s for 20 min. Lastly, imaging Homer1c and endocytic proteins fused to either mCherry or GFP was done taking images every 20 s for 10 min. In all the above-mentioned experiments Z-stacks of five to nine planes were acquired, with varying step sizes per neuron. Homer1c-mCherry was only imaged in the first and last frame. Maximum intensity images were analyzed in Fiji, by manually drawing same-size ROIs around individual puncta associated with PSDs. To measure lifetimes of clathrin and endocytic proteins we used the TrackMate plugin (*Tinevez et al., 2017*).

## Fluorescence recovery after photobleaching

FRAP experiments were performed on the spinning disk confocal system as described above, using the ILas2 system (Roche scientific). A baseline of 2 min with a 20-s interval was taken, followed by photobleaching of individual puncta with a targeted laser. The recovery of fluorescence of GFP-CLCa was imaged for 3 min with 20-s interval, followed by 12 min with 60-s interval, and 4 min with 120-s interval, resulting in a total recovery time of 19 min. For imaging the fluorescence recovery of the endocytic proteins an acquisition of 10 min was taken (2-min baseline, 3 min with 20-s interval and 7 min with 60-s interval). For acquiring FRAP images, a single Z-plane was taken. Fluorescence intensity was measured in Fiji, by manually drawing same-size ROIs around puncta. For analysis, acquisitions were corrected for drift. For each ROI, the mean intensity was measured for every time point and corrected for background and bleaching. Normalized intensities were plotted over time. Individual curves were fitted with a single-exponential function $I = A(1 – \exp^{(-Kt)})$ to estimate the mobile fraction (A) and time constant tau.

Single-molecule localization microscopy and analysis dSTORM data was acquired on the Nanoimager S from ONI (Oxford Nanoimaging Ltd.), equipped with a 100 x, 1.4NA oil immersion objective, an XYZ closed-loop piezo stage, and four laser lines: 405 nm, 471 nm, 561 nm, and 640 nm. Fluorescence emission was detected using a sCMOS camera (ORCA Flash 4, Hamamatsu). Stacks of 10,000 images were acquired at 20 Hz in TIRF mode. Samples were imaged in PBS containing 10–50 mM MEA, 5% w/v glucose, 700 µg/ml glucose oxidase, and 40 µg/ml catalase. Data was processed in NimOS software from ONI. Before each imaging session, a bead sample calibration was performed to align the two channels, achieving a channel mapping precision smaller than 8 nm. Images were rendered in ONI software and loaded into Fiji. Here, ROIs of 1 × 1 µm were drawn around individual EZs. The ROI sets were imported in Matlab (2018b) for analysis.

First, tracking was performed on the localization data to merge localizations that were detected in more than two consecutive frames as described in *Willems et al., 2020*. Next, a localization cutoff of 15 nm was taken to further analyze the localization data. A DBScan was performed to define the borders of Homer1c and CLCa in *Figure 4*, using an epsilon of 0.2 and minimum number of localizations of 100. For *Figure 6*, an epsilon of 0.35 and minimum number of localizations of 50 was used. For *Figure 6*, rings were applied to reveal the relative distribution of endocytic proteins to the EZ. Rings were calculated as a percentage of the 100% polyshape given by the DBScan. Inwards, 5 rings were created: 0–20, 20–40, 40–60, 60–80, 80–100; and outwards three rings were created: 100–120, 120–140, 140–160. Then, the number of localizations for each of the endocytic proteins

were calculated per ring. The fraction of these localizations per EZ, were plotted against the fraction of the area of the ring and normalized to 1.

## Quantification of EZ-associated synapses

For the Shank knockdown experiments, DIV14 neurons were transfected with pSM155-mCherry or pSM155-mirShank-mCherry together with Homer1c-ALFA and indicated construct. Homer1c-ALFA was labelled with JF646-conjugated FluoTAG X4 anti-ALFA (1:500 Fluotag X4, Nanotag). In the experiments manipulating actin dynamics, Homer-mCherry and GFP-CLCa expressing neurons were incubated with Latrunculin B (20 µM, Bioconnect), CK666 (400 µM, Tocris), or Jasplakinolide (20 µM, Tocris) in E4 for 30 min at 37 °C and fixed immediately after. As a control E4 containing DMSO was used. GFP-CLCa or GFP-CLCb-EED/QQN were co-expressed with Homer1c-mCherry. pSM155-mirItsn and shCPG2 were co-expressed with Homer1c-mCherry and Halo-CLCa labeled with JF646. AP2m2-WT or AP2m2-P1 was co-expressed with Homer1c-mCherry and GFP-CLCa. To quantify the fraction of synapses with an associated EZ or puncta of endocytic protein, circular regions with a fixed diameter (0.69–0.89 µm) were centered on the Homer1c signal to outline synaptic regions. These regions were then transferred to the GFP-CLC or tagged endocytic protein channel. A synapse was classified positive if the endocytic protein cluster overlapped partially or completely with the circular region. The fraction of positive synapses was calculated per cell and averaged per condition over the total population of neurons. Data plotted is normalized to the average of the control.

## Statistical analysis

Statistical significance was tested using a Student's t-test when comparing two groups. When comparing multiple groups statistical significance was tested using a one-way ANOVA followed by a Tukey or Dunnett's multiple comparison post-hoc test. All the statistical tests with a p-value below 0.05 were considered significant. In all figures, significance is indicated as follows: $p < 0.05$ is indicated by *, $p < 0.01$ by **, and $p < 0.001$ by ***. Analysis was performed on neurons originating from at least two individual batches of hippocampal neurons. Number of neurons used for analysis is indicated as $N$, number of spines or CCSs is represented as $n$.

## Acknowledgements

We thank all members of the MacGillavry lab for support and discussions. This work was supported by the Netherlands Organization of Scientific Research (NWO-ALWOP. 191 to HDM).

## Additional information

### Funding

| Funder | Grant reference number | Author |
| --- | --- | --- |
| Nederlandse Organisatie voor Wetenschappelijk Onderzoek | ALW Open Program grant NWO-ALWOP. 191 | Harold D MacGillavry |

The funders had no role in study design, data collection and interpretation, or the decision to submit the work for publication.

### Author contributions

Lisa AE Catsburg, Conceptualization, Formal analysis, Investigation, Methodology, Validation, Visualization, Writing - original draft, Writing - review and editing; Manon Westra, Formal analysis, Investigation, Methodology, Software, Validation, Writing - review and editing; Annemarie ML van Schaik, Formal analysis, Investigation, Methodology, Validation, Visualization, Writing - review and editing; Harold D MacGillavry, Conceptualization, Formal analysis, Funding acquisition, Investigation, Methodology, Resources, Supervision, Validation, Visualization, Writing - original draft, Writing - review and editing

## Author ORCIDs
Lisa AE Catsburg http://orcid.org/0000-0001-6870-3149
Manon Westra http://orcid.org/0000-0003-4027-1115
Harold D MacGillavry http://orcid.org/0000-0002-6153-3586

## Ethics

All animal experiments were performed in compliance with the guidelines for the welfare of experimental animals issued by the Government of the Netherlands (Wet op de Dierproeven, 1996) and European regulations (Guideline 86/609/EEC). All animal experiments were approved by the Dutch Animal Experiments Review Committee (Dier Experimenten Commissie; DEC), performed in line with the institutional guidelines of Utrecht University.

## Decision letter and Author response

Decision letter https://doi.org/10.7554/eLife.74387.sa1
Author response https://doi.org/10.7554/eLife.74387.sa2

---

# Additional files

## Supplementary files
• Transparent reporting form

## Data availability

All relevant data are within the paper and its Supporting Information files. All the numerical data that are represented as a graph in a figure are provided in the Source Data file.

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
