## [Editor Report]

The delineation of protein organization in the perisynaptic endocytic zone is an important contribution to our understanding of synapse structure, and new observations about changes to this structure establish intriguing new phenomenology that appears closely linked to synapse functional plasticity. Cutting-edge genetic tagging and elegant application of super-resolution imaging compellingly support the key claims in the paper.

---

## [Decision Letter]

**Decision letter after peer review:**

[Editors’ note: the authors submitted for reconsideration following the decision after peer review. What follows is the decision letter after the first round of review.]

Thank you for submitting your work entitled "Dynamics and nanoscale organization of the postsynaptic endocytic zone at excitatory synapses" for consideration by *eLife*. Your article has been reviewed by 3 peer reviewers, one of whom is a member of our Board of Reviewing Editors, and the evaluation has been overseen by a Senior Editor. The reviewers have opted to remain anonymous.

We are sorry to say that, after consultation with the reviewers, we have decided that your work will not be considered further for publication by *eLife*.

The reviewers find that the manuscript is largely descriptive and lacks mechanistic insights. However, there was a consensus that the manuscript could be improved by tagging a selected subset of endogenous most relevant proteins of the complex and examining the effects of neuronal activity or plasticity-inducing stimuli on the kinetics of the EZ proteins. Therefore, we would like to open the possibility that you address these issues and re-submit the manuscript.

*Reviewer #1:*

The authors characterize the molecular organization of the endocytic zone (EZ) known to associate PSD with clathrin coats using advanced imaging approaches, including high-resolution STED imaging and time-lapse live-cell imaging. They also test 12 well-known endocytic proteins to see whether and how they colocalize with EZ and find that a subset of them show strong and long-lived colocalizations, whereas others show weak and transient colocalizations, with EZ by advanced imaging and cell biological disruptions. They find Shank is a key component of the postsynaptic density that coordinates the targeting and localization of the new EZ components. This is a careful and comprehensive analysis of the molecular composition and dynamics of known and additional EZ components using high-resolution and time-lapse imaging. The results are largely convincing and conclusive and provide insights into how synaptic membrane proteins and receptors are trafficked in and out of synapses to regulated synaptic plasticity, which would have significant impacts on broad fields of molecular and cellular neuroscience.

The authors test many well-known endocytic proteins for EZ colocalization using GFP tagging, but it is unclear to me whether these taggings might affect the stability of the proteins, normal distribution of the proteins to the EZ, or any normal protein-protein interactions, perhaps misleading the conclusions.

*Reviewer #2:*

Catsburg et al. attack an interesting topic with a combination of advanced molecular and imaging approaches. The endocytic zone in neuronal spines congregates the endocytic protein clathrin near synapses, and has been demonstrated in several studies to be of functional importance for regulating synaptic transmission by influencing the endocytosis and recycling of glutamate receptors. Basic characterization of this "zone" has been provided by these authors and others, but a more complete description of its character or components has been essentially missing in the field.

The work first provides new descriptive characterization of clathrin structures in spines vs dendrites using an excellent knock-in approach they recently developed. These results are straightforward but important validation of earlier findings here using the CRISPR approach, and also will be useful baseline information with which to examine changes in the zone for instance during various forms of neuronal plasticity. Then, taking what seems like a tip from the work in other cells as to the organization of proteins at sites of clathrin-mediated endocytosis, they survey the abundance and stability vs transience of a large set of relevant proteins. This is fundamental information necessary to discern the nature and role of the zone.

The authors then test two mechanisms that might hold the zone or its proteins in place, connections to the PSD protein Shank or to the abundant spine actin cytoskeleton. Experiments to manipulate these two connections do not demonstrate unraveling of the entire system, but instead show a fairly remarkable and specific loss of different proteins. Most notably (to this reader), after Shank knockdown, beta2 adaptin (part of the AP2 complex that was originally used to define the zone in EM) is lost, but Dyn2 (a cytosolic and dynamically recruited part of the machinery during CME) is retained. These and the related results provide new insight to the complex mechanisms that must be at work governing the assembly and dynamics of this specialized domain in spines.

Overall, the work is systematically and rigorously conducted, and the results are nicely presented with a minor few exceptions. The knock-in application will help establish this general approach as the new standard in the field. Indeed, the overexpression approach used for the protein survey by contrast with the elegant knock-ins feels slightly disappointing, indicating how important it is that the field move as quickly as possible to utilize knock-ins where possible. The overall conclusions about the components, dynamics, and mechanisms of the endocytic zone are stated carefully, and add greatly to understanding of this structure.

Major conclusions (e.g. distribution of a key protein or two, or retention of Dyn2 but not adaptin or Eps15 after Shank knockdown) could be confirmed via immunocytochemistry if the available antibodies are suitable.

CPG2 is an interesting case, since it is very stable and the authors conclude it appears to be integral to the endocytic zone. This interesting conclusion would be strengthened by including its analysis in the spatial distribution in Figure 6 and the knockdown experiments of Figure 7.

It would be good to verify that the CLC knock-in does not alter total clathrin content in the neurons, and also that it does not affect clathrin-mediated endocytosis, though the abundant literature might provide sufficient evidence if this latter point has been tested directly.

Line 214 and Figure 3c: how can the EZ centroid be within the PSD? Are these structures overlapped in Z, or is there another explanation?

The distribution of proteins documented in Figure 6G is certainly interesting, but fairly hard to interpret; the differences in the graphs are fairly pointed out in the text, but not readily apparent without close investigation and reading. The graph in 6F is not a very helpful way to compare this set of proteins, and either a different LUT or a traditional line graph overlay might be better here.

The functional impact of the current findings is left for the most part unexplored, probably because this seems extremely difficult to attack. However, if the authors had the ability to examine any of the parameters measured here (protein abundance, structural characteristics of the zone, protein mobility) in the context of common neuron plasticity paradigms, it would likely fill that gap. To my reading, this is not necessary.

The discussion is rather long, and some of the speculation there could be made more concise.

Line 115: what is the STED resolution achieved, and is it capable of resolving the differences seen in previous EM studies? Speculation about the nature of the small structures in static images surely may be limited by the resolution.

The close apposition of PSD and endocytic zone suggests that the authors might be able to examine published synaptosome proteomes to see whether their observations of protein abundance are reflected in those datasets.

Figure 7C: what are the arrows pointing to? It looks like the one labeled "Shank homer Dyn3" is pointing at actin.

*Reviewer #3:*

In this manuscript, the authors first reproduce with some additional details, in particular using various superresolution approaches, our knowledge of the peri-synaptic localization of clathrin labelled endocytic zones (EZ) and determine that peri-synaptic EZ are dynamically distinct from shaft clathrin-coated structures, being more stable. They then perform an extensive characterization of the localization of a set of endocytic accessory proteins and show that a large fraction localizes to the perisynapse. These include HIP1R, ꞵ2-adaptin, Dyn2, CPG2, Eps15, and Itsn1L. In contrast, a subset of other endocytic proteins (PICALM, Epsn2, Amph and FCHO1) were less enriched at EZ. With time lapse video-microscopy, they analyze the residence of these various endocytic proteins next to the PSD and find that they exhibit a range of behavior, from short lived proteins such as FCHO1 to proteins more stably associated to the PSD.

Next, using two-color single molecule localization microscopy, the authors study the colocalization of these various proteins with respect to clathrin in EZ, and find that endocytic proteins have distinct spatial organization relative to the clathrin structure marking the EZ. They find that ꞵ2-adaptin, Eps15, and Itsn1L were often distributed in smaller patches around and sometimes within the EZ. HIP1R showed a more homogenous distribution and often colocalized with the EZ. Dyn2 showed an overall more homogenous distribution, similar to HIP1R.

Finally, they touch upon the mechanism of EZ localization next to the PSD. They first recapitulate the central role of Shank by finding that Shank-KD also reduced the association of the PSD with other endocytic proteins in addition to clathrin. They then use a pharmacological approach to suggest that EZ positioning is not related to actin dynamics, and overexpressed an AP2 mutant that unable to interact with PIP2 to suggest that interaction with the membrane is not necessary for EZ positioning.

Altogether this is a carefully performed study, but the knowledge acquired remains relatively incremental to our previous understanding of EZ positioning.

The identification of proteins of the endocytic machinery localized at EZ in not surprising and lacks mechanical insight. The study of the mechanism of EZ positioning next to the PSD also lacks additional insight compared to our previous knowledge of the implication of Shank. Finally, this study does not address either the mechanism of EZ assembly nor its function, nor its regulation, which are to my sense the important questions to answer regarding EZs.

Several lines could be developed to improve the impact of this study.

The authors could have for example studied the role of the various components in EZ function through coated pit formation, or cargo internalization. They could also have analyzed the impact of neuronal activity on the recruitment of the various components.

Regarding the experimental approach, several experiments rely of overexpression of tagged components which can have unpredictable effects on their localization. The authors elegantly performed gene editing of clathrin light chain to verify its localization at endogenous levels. It would have been nice to apply the same approach for the study of the other components.

---

## [Author Response]

[Editors’ note: the authors resubmitted a revised version of the paper for consideration. What follows is the authors’ response to the first round of review.]

Reviewer #1:The authors characterize the molecular organization of the endocytic zone (EZ) known to associate PSD with clathrin coats using advanced imaging approaches, including high-resolution STED imaging and time-lapse live-cell imaging. They also test 12 well-known endocytic proteins to see whether and how they colocalize with EZ and find that a subset of them show strong and long-lived colocalizations, whereas others show weak and transient colocalizations, with EZ by advanced imaging and cell biological disruptions. They find Shank is a key component of the postsynaptic density that coordinates the targeting and localization of the new EZ components. This is a careful and comprehensive analysis of the molecular composition and dynamics of known and additional EZ components using high-resolution and time-lapse imaging. The results are largely convincing and conclusive and provide insights into how synaptic membrane proteins and receptors are trafficked in and out of synapses to regulated synaptic plasticity, which would have significant impacts on broad fields of molecular and cellular neuroscience.

We thank the reviewer for the encouraging evaluation of our manuscript.

The authors test many well-known endocytic proteins for EZ colocalization using GFP tagging, but it is unclear to me whether these taggings might affect the stability of the proteins, normal distribution of the proteins to the EZ, or any normal protein-protein interactions, perhaps misleading the conclusions.

Overexpression of tagged proteins can certainly have adverse effects on protein stability, distribution or function. Nevertheless, to compare the distribution and live-cell dynamics of these proteins, at the start of this project we decided to use tagged exogenous proteins as this is still the most direct, robust and broadly accepted approach in the field (and our knockin approach was still being developed). As we are aware of these caveats, throughout the experiments we ensured that only neurons with modest expression levels were selected for analysis. To address this point more directly, we took advantage of the recently established CRISPR/Cas9 approach in the lab (Willems *et al.,* PLOS Biology 2020), and developed knockin constructs for a number of key endocytic proteins. Unfortunately, despite testing several knockin approaches to tag endogenous AP2 subunits, we could not reliably tag the endogenous AP2 complex. Nevertheless, we successfully developed knockin constructs for other key endocytic proteins and found that endogenously tagged Eps15, Itsn1 and Dyn2 were distributed similar as we observed for exogenously tagged proteins. Moreover, we optimized antibody staining protocols for these proteins and could confirm that also endogenously labeled proteins were similarly distributed, further confirming that overexpression or tagging of these proteins did not disrupt their distribution. These results are now included in the revised manuscript and described on page 13.

Reviewer #2:Catsburg et al. attack an interesting topic with a combination of advanced molecular and imaging approaches. The endocytic zone in neuronal spines congregates the endocytic protein clathrin near synapses, and has been demonstrated in several studies to be of functional importance for regulating synaptic transmission by influencing the endocytosis and recycling of glutamate receptors. Basic characterization of this "zone" has been provided by these authors and others, but a more complete description of its character or components has been essentially missing in the field.The work first provides new descriptive characterization of clathrin structures in spines vs dendrites using an excellent knock-in approach they recently developed. These results are straightforward but important validation of earlier findings here using the CRISPR approach, and also will be useful baseline information with which to examine changes in the zone for instance during various forms of neuronal plasticity. Then, taking what seems like a tip from the work in other cells as to the organization of proteins at sites of clathrin-mediated endocytosis, they survey the abundance and stability vs transience of a large set of relevant proteins. This is fundamental information necessary to discern the nature and role of the zone.The authors then test two mechanisms that might hold the zone or its proteins in place, connections to the PSD protein Shank or to the abundant spine actin cytoskeleton. Experiments to manipulate these two connections do not demonstrate unraveling of the entire system, but instead show a fairly remarkable and specific loss of different proteins. Most notably (to this reader), after Shank knockdown, beta2 adaptin (part of the AP2 complex that was originally used to define the zone in EM) is lost, but Dyn2 (a cytosolic and dynamically recruited part of the machinery during CME) is retained. These and the related results provide new insight to the complex mechanisms that must be at work governing the assembly and dynamics of this specialized domain in spines.Overall, the work is systematically and rigorously conducted, and the results are nicely presented with a minor few exceptions. The knock-in application will help establish this general approach as the new standard in the field. Indeed, the overexpression approach used for the protein survey by contrast with the elegant knock-ins feels slightly disappointing, indicating how important it is that the field move as quickly as possible to utilize knock-ins where possible. The overall conclusions about the components, dynamics, and mechanisms of the endocytic zone are stated carefully, and add greatly to understanding of this structure.

We thank the reviewer for the careful analysis of our manuscript, the raised points have encouraged us to further characterize endogenous distribution of key endocytic proteins

Major conclusions (e.g. distribution of a key protein or two, or retention of Dyn2 but not adaptin or Eps15 after Shank knockdown) could be confirmed via immunocytochemistry if the available antibodies are suitable.

We were successful in labeling these proteins with immunohistochemistry and confirmed that the distribution is consistent with the findings we present using tagged expressed proteins and CRISPR/Cas9-mediated genome editing, see also our response to reviewer 1 and 3. However, as immunohistochemistry labels all neurons in our dense cultures, the analysis of PSD-associated puncta specifically in transfected neurons (e.g., with the Shank knockdown construct) is highly compromised and could unfortunately not be performed in a reliable manner.

CPG2 is an interesting case, since it is very stable and the authors conclude it appears to be integral to the endocytic zone. This interesting conclusion would be strengthened by including its analysis in the spatial distribution in Figure 6 and the knockdown experiments of Figure 7.

We agree CPG2 is an interesting candidate protein and based on previous work from the Nedivi lab and our findings we hypothesized it could fulfil a key role as stabilizing scaffold in the EZ. We therefore tested whether shRNA-mediated knockdown of endogenous CPG2 destabilizes the EZ. However, we found no difference in PSD-EZ association in CPG2knockdown neurons. These results are included in the revised manuscript and described on page 21 and Figure 7. We attempted to analyze the spatial distribution of CPG2 using SMLM as presented in Figure 6 for other endocytic proteins, but we were unfortunately unable to collect reliable datasets as the localization densities acquired for CPG2 were not sufficiently high, preventing accurate estimates of single-molecule distribution.

It would be good to verify that the CLC knock-in does not alter total clathrin content in the neurons, and also that it does not affect clathrin-mediated endocytosis, though the abundant literature might provide sufficient evidence if this latter point has been tested directly.

Tagging clathrin light-chain (CLC) at the N-terminus has been a standard and wellcharacterized approach in the field to tag clathrin structures in live cells as it does not affect clathrin-mediated endocytosis (Engqvist-Goldstein *et al.,* JCB 2001; Gaidarov *et al.,* Nat Cell Biology 1999; Wu *et al.,* JCB 2001), also in neurons (Blanpied *et al.,* Neuron 2002). We therefore designed the clathrin light-chain knock-in construct such that it is similar to the overexpression construct. Unfortunately, precise quantification of clathrin levels in knockin neurons is difficult. Determining protein levels using Western blot analysis is not feasible as only a few cells (< 1% of the culture) are positive for GFP-CLC. Also, in our hands, the quality of attempts to label with commercial antibodies has been too low to reliably detect absolute protein levels in individual cells. We do observe that the levels of endogenous GFP-CLC are sufficient to readily detect in individual cells and are comparable between cells, indicating that the knockin approach does not induce undesired indel mutations, consistent with our deep-sequencing analysis of donor integration in knockin neurons (Willems *et al.,* PLOS Biology 2020).

Line 214 and Figure 3c: how can the EZ centroid be within the PSD? Are these structures overlapped in Z, or is there another explanation?

We performed 2D-STORM imaging in these experiments, so this is indeed most likely reflecting overlap in Z.

The distribution of proteins documented in Figure 6G is certainly interesting, but fairly hard to interpret; the differences in the graphs are fairly pointed out in the text, but not readily apparent without close investigation and reading. The graph in 6F is not a very helpful way to compare this set of proteins, and either a different LUT or a traditional line graph overlay might be better here.

We regret the presentation of the data is hard to interpret. We have compared several ways to present the different aspects of information that we could extract from these experiments. As to the reviewer’s suggestion, we replaced the heatmap for a histogram line graph (Figure 6F). We also added three example maps showing how the rings are drawn around the EZ (Figure 6G). We hope this will help convey the principle of the ring analysis.

The functional impact of the current findings is left for the most part unexplored, probably because this seems extremely difficult to attack. However, if the authors had the ability to examine any of the parameters measured here (protein abundance, structural characteristics of the zone, protein mobility) in the context of common neuron plasticity paradigms, it would likely fill that gap. To my reading, this is not necessary.

We undertook experiments to address how synaptic plasticity-inducing stimuli alter the structure of the EZ that we think add functional impact to our findings. Strikingly, we found that mGluR-LTD, but not NMDAR-LTD induced a rapid, but transient increase in the number of clathrin-coated structures associated with synapses. Moreover, we found that a chemical LTP protocol induced a long-lasting increase in the number of PSDs associated with multiple EZs. This latter finding, we followed up with live-cell STED imaging, providing for the first-time evidence that potentiation of synapses is accompanied with the molecular reorganization of the perisynaptic endocytic machinery. We present these data in Figure 8 and discuss these on page 24 and 25.

The discussion is rather long, and some of the speculation there could be made more concise.

We shortened the discussion considerably and formulated our speculations more concisely. We think the discussion is now more to the point.

Line 115: what is the STED resolution achieved, and is it capable of resolving the differences seen in previous EM studies? Speculation about the nature of the small structures in static images surely may be limited by the resolution.

The resolution we typically gain with STED is 50-100 nm, indeed far from the nanometer resolution provided by EM. We here intended to refer to similar analyses performed in the cited papers using confocal microscopy (Saffarian *et al.,* 2009) and other super-resolution studies (Grove *et al.,* 2014; Figure 2 and Leyton-Puig *et al.,* 2017; Figure 1c). We adapted our conclusion to include the notion that perhaps the imaging resolution is not sufficient to make the distinction between clathrin pits and lattices.

The close apposition of PSD and endocytic zone suggests that the authors might be able to examine published synaptosome proteomes to see whether their observations of protein abundance are reflected in those datasets.

This is a very interesting suggestion by the reviewer, indeed many endocytic proteins are identified in proteomics studies on synaptosomes (e.g., see Bieseman *et al.,* EMBO J 2014). In such preparations there is however a contribution of both the presynaptic and postsynaptic compartments. Since there is in fact a strong accumulation of similar endocytic proteins (e.g., clathrin, dynamin, Itsn1, AP2) at presynaptic sites, quantifications of relative abundance will be difficult to interpret.

Figure 7C: what are the arrows pointing to? It looks like the one labeled "Shank homer Dyn3" is pointing at actin.

We intended to illustrate that the EZ is coupled to the PSD via Shank-Homer-Dyn3 interactions, however we agree this is not clear. We removed the arrows in this illustration.

Reviewer #3:In this manuscript, the authors first reproduce with some additional details, in particular using various superresolution approaches, our knowledge of the peri-synaptic localization of clathrin labelled endocytic zones (EZ) and determine that peri-synaptic EZ are dynamically distinct from shaft clathrin-coated structures, being more stable. They then perform an extensive characterization of the localization of a set of endocytic accessory proteins and show that a large fraction localizes to the perisynapse. These include HIP1R, ꞵ2-adaptin, Dyn2, CPG2, Eps15, and Itsn1L. In contrast, a subset of other endocytic proteins (PICALM, Epsn2, Amph and FCHO1) were less enriched at EZ. With time lapse video-microscopy, they analyze the residence of these various endocytic proteins next to the PSD and find that they exhibit a range of behavior, from short lived proteins such as FCHO1 to proteins more stably associated to the PSD.Next, using two-color single molecule localization microscopy, the authors study the colocalization of these various proteins with respect to clathrin in EZ, and find that endocytic proteins have distinct spatial organization relative to the clathrin structure marking the EZ. They find that ꞵ2-adaptin, Eps15, and Itsn1L were often distributed in smaller patches around and sometimes within the EZ. HIP1R showed a more homogenous distribution and often colocalized with the EZ. Dyn2 showed an overall more homogenous distribution, similar to HIP1R.Finally, they touch upon the mechanism of EZ localization next to the PSD. They first recapitulate the central role of Shank by finding that Shank-KD also reduced the association of the PSD with other endocytic proteins in addition to clathrin. They then use a pharmacological approach to suggest that EZ positioning is not related to actin dynamics, and overexpressed an AP2 mutant that unable to interact with PIP2 to suggest that interaction with the membrane is not necessary for EZ positioning.Altogether this is a carefully performed study, but the knowledge acquired remains relatively incremental to our previous understanding of EZ positioning.The identification of proteins of the endocytic machinery localized at EZ in not surprising and lacks mechanical insight. The study of the mechanism of EZ positioning next to the PSD also lacks additional insight compared to our previous knowledge of the implication of Shank. Finally, this study does not address either the mechanism of EZ assembly nor its function, nor its regulation, which are to my sense the important questions to answer regarding EZs.Several lines could be developed to improve the impact of this study.The authors could have for example studied the role of the various components in EZ function through coated pit formation, or cargo internalization. They could also have analyzed the impact of neuronal activity on the recruitment of the various components.

We are motivated to track the dynamic behavior of the EZ and components through different phases of endocytosis. However, the specific and efficient labeling of cargoes (e.g., synaptic receptors) that undergo endocytosis is not trivial. Thus, correlating recruitment of EZ components to coated pit formation or cargo internalization, or gaining detailed mechanistic insight in the contribution of individual components to this process is technically highly challenging – as acknowledged by reviewer 2. Moreover, as we and others found that many components of the EZ are stably enriched and appear ‘optically stable’, visualization of endocytosis by tracking the appearance and disappearance of markers that are frequently used in other cell types, such as clathrin or dynamin, is unfortunately not feasible at the EZ. Instead, we decided to study the impact of synaptic plasticity paradigms on the structural organization of the EZ.

Regarding the experimental approach, several experiments rely of overexpression of tagged components which can have unpredictable effects on their localization. The authors elegantly performed gene editing of clathrin light chain to verify its localization at endogenous levels. It would have been nice to apply the same approach for the study of the other components.

We agree this is an important point, and this point has also been raised by the other reviewers. We developed knockin constructs for a number of key endocytic proteins and found that the distribution of endogenously tagged Eps15, Itsn1 and Dyn2 was not different than as we observed for exogenously tagged proteins. Moreover, we used immunostaining for Eps15, Itsn1 and Dyn2 to further confirm that also endogenously labeled proteins were similarly distributed. We therefore concluded that overexpression or tagging of these proteins did not disrupt their distribution. These results are now included in the revised manuscript and described on page 13.